# On the Inductive Bias of a CNN for Distributions with Orthogonal Patterns

## Abstract

Training overparameterized convolutional neural networks with gradient based optimization is the most successful learning method for image classification. However, their generalization properties are far from understood. In this work, we consider a simplified image classification task where images contain orthogonal patches and are learned with a 3-layer overparameterized convolutional network and stochastic gradient descent (SGD). We empirically identify a novel phenomenon of SGD in our setting, where the dot-product between the learned pattern detectors and their detected patterns are governed by the pattern statistics in the training set. We call this phenomenon Pattern Statistics Inductive Bias (PSI) and empirically verify it in a large number of instances. We prove that in our setting, if a learning algorithm satisfies PSI then its sample complexity is $O(d^2 \log(d))$ where $d$ is the filter dimension. In contrast, we show a VC dimension lower bound which is exponential in $d$. We perform experiments with overparameterized CNNs on a variant of MNIST with non-orthogonal patches, and show that the empirical observations are in line with our analysis.

## 1 Introduction

Convolutional neural networks (CNNs) have achieved remarkable performance in various computer vision tasks (Krizhevsky et al., 2012; Xu et al., 2015; Taigman et al., 2014). In practice, these networks typically have more parameters than needed to achieve zero train error (i.e., are overparameterized). Despite non-convexity and the potential problem of overfitting, training these models with gradient based methods leads to solutions with low test error. It is still largely unknown why such simple optimization algorithms have outstanding test performance for learning overparameterized convolutional networks.

Recently, there have been major efforts to provide generalization guarantees for overparameterized CNNs. However, current generalization guarantees either depend on the number of channels of the network (Long & Sedghi, 2020) or hold under specific constraints on the weights (Li et al., 2018).

Clearly, the generalization of overparameterized CNNs depends on both the learning algorithm (gradient-based methods) and unique properties of the data. Providing generalization guarantees while incorporating these factors is a major challenge. Indeed, this requires analyzing non-convex optimization methods and mathematically defining properties of the data, which is extremely difficult for real-world problems. Therefore, it is necessary to first understand simple settings which are amenable to theoretical and empirical analysis and share salient features with real-world problems.

Towards this goal, we analyze a simplified pattern recognition task where all patterns in the images are orthogonal and the classification is binary. The architecture is a 3-layer overparameterized convolutional neural network and it is learned using stochastic gradient descent (SGD). We take a unique approach that combines novel empirical observations with theoretical guarantees to provide a novel generalization bound which is independent of the number of channels and is a low-degree polynomial of the filter dimension, which is usually low in practice.

Empirically, we identify a novel property of the solutions found by SGD. We observe that the statistics of patterns in the training data govern the magnitude of the dot-product between learned pattern detectors and their detected patterns. Specifically, patterns that appear almost exclusively in one of the classes will have a large dot-product with the channels that detect them. On the other hand,

patterns that appear roughly equally in both classes, will have a low dot-product with their detecting channels. We formally define this as the "Pattern Statistics Inductive Bias" condition (PSI) and provide empirical evidence that PSI holds across a large number of instances. We also prove that SGD indeed satisfies PSI in a simple setup of two points in the training set.

Under the assumption that PSI holds, we analyze the sample complexity and prove that it is at most $O(d^2 \log d)$, where $d$ is the filter dimension. In contrast, we show that the VC dimension of the class of functions we consider is exponential in $d$, and thus there exist other learning algorithms (not SGD) that will have exponential sample complexity. Together, these results provide firm evidence that even though SGD can in principle overfit, it is nonetheless biased towards solutions which are determined by the statistics of the patterns in the training set and consequently it has good generalization performance.

We perform experiments with overparamterized CNNs on a variant of MNIST that has **non-orthogonal** patterns. We use our analysis to better understand why SGD has low sample complexity in this setting. We empirically show that the inductive bias of SGD is similar to PSI. This suggests that the idea of PSI is not unique to the orthogonal case and can be useful for understanding overparameterized CNNs in other challenging settings.

## 2 RELATED WORK

Several recent works have studied the generalization properties of overparameterized CNNs. Some of these propose generalization bounds that depend on the number of channels (Long & Sedghi, 2020; Jiang et al., 2019). Others provide guarantees for CNNs with constraints on the weights (Zhou & Feng, 2018; Li et al., 2018). Convergence of gradient descent to KKT points of the max-margin problem is shown in Lyu & Li (2020) and Nacson et al. (2019) for homogeneous models. However, their results do not provide generalization guarantees in our setting. Gunasekar et al. (2018) study the inductive bias of linear CNNs.

Yu et al. (2019) study a pattern classification problem similar to ours. However, their analysis holds for an unbounded hinge loss which is not used in practice. Furthermore, their sample complexity depends on the network size, and thus does not explain why large networks do not overfit. Other works have studied learning under certain ground truth distributions. For example, Brutzkus & Globerson (2019) study a simple extension of the XOR problem, showing that overparameterized CNNs generalize better than smaller CNNs. Single-channel CNNs are analyzed in (Du et al., 2018b;a; Brutzkus & Globerson, 2017; Du et al., 2018c).

Other works study the inductive bias of gradient descent on fully connected linear or non-linear networks (Ji & Telgarsky, 2019; Arora et al., 2019a; Wei et al., 2019; Brutzkus et al., 2018; Dziugaite & Roy, 2017; Allen-Zhu et al., 2019; Chizat & Bach, 2020). Fully connected networks were also analyzed via the NTK approximation (Du et al., 2019; 2018d; Arora et al., 2019b; Fiat et al., 2019). Kushilevitz & Roth (1996); Shvaytser (1990) study the learnability of visual patterns distribution. However, our focus is on learnability using a specific algorithm and architecture: SGD trained on overparameterized CNNs.

## 3 THE ORTHOGONAL PATTERNS PROBLEM

**Data Generating Distribution:** We consider a learning problem that captures a key property of visual classification. Many visual classes are characterized by the existence of certain patterns. For example an 8 will typically contain an x like pattern somewhere in the image. Here we consider an abstraction of this behavior where images consist of a set of patterns. Furthermore, each class is characterized by a pattern that appears exclusively in it. We define this formally below.

Let $\mathcal{P}$ be a set of orthogonal vectors in $\mathbb{R}^d$, where $|\mathcal{P}| \leq d$. For simplicity, we assume that $\|\boldsymbol{p}\|_2 = 1$ for all $\boldsymbol{p} \in \mathcal{P}$. We consider input vectors $\boldsymbol{x}$ with $n$ patterns of dimension $d$. Formally, $\boldsymbol{x} = (\boldsymbol{x}[1], ..., \boldsymbol{x}[n]) \in \mathbb{R}^{nd}$ where $\boldsymbol{x}[i] \in \mathcal{P}$ is the $i$th pattern of $\boldsymbol{x}$ and $n < d$. We denote $\boldsymbol{p} \in \boldsymbol{x}$ if $\boldsymbol{x}$ contains the pattern $\boldsymbol{p} \in \mathcal{P}$.[1] Let $\mathcal{P}(\boldsymbol{x}) = \{\boldsymbol{p} \in \boldsymbol{x} \mid \boldsymbol{p} \in \mathcal{P}\}$ denote the set of all patterns in $\boldsymbol{x}$.

---

[1] We say that $\boldsymbol{x}$ contains $\boldsymbol{p}$ if there exists $j$ such that $\boldsymbol{x}[j] = \boldsymbol{p}$.

Next, we define how labeled points are generated. Consider three non-overlapping sets of patterns: $\mathcal{P}_-, \mathcal{P}_+, \mathcal{P}_s \subset \mathcal{P}$ whose disjoint union is $\mathcal{P}$. $\mathcal{P}_+$ is the set of positive patterns, $\mathcal{P}_-$ the set of negative patterns and $\mathcal{P}_s$ is the set of spurious patterns. For simplicity, in this work we consider the case where $|\mathcal{P}_+| = |\mathcal{P}_-| = 1$. We denote, $\mathcal{P} = \{\boldsymbol{p}_1, \boldsymbol{p}_2, ..., \boldsymbol{p}_{|\mathcal{P}|}\}$, $\mathcal{P}_+ = \{\boldsymbol{p}_1\}$ and $\mathcal{P}_- = \{\boldsymbol{p}_2\}$. For convenience, we also refer to a set of patterns $A$ as a set of the indices of the patterns, .e.g., we denote $i \in A$ if $\boldsymbol{p}_i \in A$.

We consider distributions $\mathcal{D}$ over $(\boldsymbol{x}, y) \in \mathbb{R}^{nd} \times \{\pm 1\}$ with the following properties: (1) $\mathbb{P}(y = 1) = \mathbb{P}(y = -1) = \frac{1}{2}$. (2) Given $y = 1$, a vector $\boldsymbol{x}$ is sampled as follows. Choose the positive pattern $\boldsymbol{p}_1$ and randomly choose a set of $n - 1$ patterns from $\mathcal{P}_s$. Denote this set of $n$ chosen patterns by $A$. Let $\boldsymbol{x}$ be some $\boldsymbol{x}'$ such that $\mathcal{P}(\boldsymbol{x}') = A$, i.e., the location of each pattern in $\boldsymbol{x}$ is chosen arbitrarily.[2] For example, if $n = 3$ and the $n - 1$ patterns are $\boldsymbol{p}_3, \boldsymbol{p}_7$ this can result in samples such as $([\boldsymbol{p}_1, \boldsymbol{p}_7, \boldsymbol{p}_3], 1)$ or $([\boldsymbol{p}_3, \boldsymbol{p}_1, \boldsymbol{p}_7], 1)$. (3) Similarly for $y = -1$, only choose $\boldsymbol{p}_2 \in \mathcal{P}_-$ instead of $\boldsymbol{p}_1$.

We will consider several distributions that satisfy the above and have different sampling schemes for the spurious patterns (see Sec. 7). Fig. 4 in the supplementary shows an example of samples generated using the above procedure. Note that any distribution which satisfies the above is linearly separable and each vector $\boldsymbol{x}$ can be classified solely based on whether $\boldsymbol{p}_1 \in \boldsymbol{x}$ or $\boldsymbol{p}_2 \in \boldsymbol{x}$.

**Neural Architecture:** For learning the above pattern detection problems, a natural model in this context is a 3-layer network with a convolutional layer, followed by ReLU, max pooling and a fully-connected layer. Each channel in the first layer can be thought of as a detector for a given pattern. We say that a detector detects pattern $\boldsymbol{p} \in \mathcal{P}$, if $\boldsymbol{p}$ has the largest dot product with the detector among all patterns in $\mathcal{P}_+ \cup \mathcal{P}_s$ or $\mathcal{P}_- \cup \mathcal{P}_s$ and this dot product is positive.[3] For simplicity we fix the weights on the last linear layer to values $\pm 1$.[4]

Let $2k$ denote the number of channels. We partition the channels into two sets: $\boldsymbol{w}^{(1)}, \ldots, \boldsymbol{w}^{(k)}$ and $\boldsymbol{u}^{(1)}, \ldots, \boldsymbol{u}^{(k)}$. These will have weights of $+1$ and $-1$ in the output respectively. Finally, let $W \in \mathbb{R}^{2k \times 2}$ be the weight matrix whose rows are $\boldsymbol{w}^{(i)}$ followed by $\boldsymbol{u}^{(i)}$.

For an input $\boldsymbol{x} = (\boldsymbol{x}[1], ..., \boldsymbol{x}[n]) \in \mathbb{R}^{nd}$ where $\boldsymbol{x}[i] \in \mathbb{R}^d$, the output of the network is:

$$N_W(\boldsymbol{x}) = \sum_{i=1}^{k} \left[ \max_j \left\{ \sigma\left(\boldsymbol{w}^{(i)} \cdot \boldsymbol{x}[j]\right) \right\} - \max_j \left\{ \sigma\left(\boldsymbol{u}^{(i)} \cdot \boldsymbol{x}[j]\right) \right\} \right] \tag{1}$$

where $\sigma(x) = \max\{0, x\}$ is the ReLU activation. Let $\mathcal{H}$ denote the class of all networks $N_W$ in Eq. 1, with $k > 0$. Finally, we note that $\mathcal{H}$ can perfectly fit the distribution $\mathcal{D}$ above, by setting $k = 1$, $\boldsymbol{w}^{(1)} = \boldsymbol{p}_1$ and $\boldsymbol{u}^{(1)} = \boldsymbol{p}_2$. Therefore, for $k > 1$ the network is overparameterized.

**Training Algorithm:** Let $S$ be a training set with $m$ IID samples from $\mathcal{D}$. We consider minimizing the hinge loss: $\ell(W) = \frac{1}{m} \sum_{(\boldsymbol{x}_i, y_i) \in S} \max\{1 - y_i N_W(\boldsymbol{x}_i), 0\}$. For optimization, we use SGD with constant learning rate $\eta$. The parameters $W$ are initialized as IID Gaussians with zero mean and standard deviation $\sigma_g$. Let $W_t$ be the weight matrix at iteration $t$ of SGD. Similarly let $\boldsymbol{w}_t^{(i)}, \boldsymbol{u}_t^{(i)}$ be the corresponding vectors at iteration $t$.

**Detection Ratios:** We now define the notion of detection ratios. The detection ratios are a property of the model and will be key to our analysis. We first define the set of neurons that are maximally activated by pattern $\boldsymbol{p}_i$ among all patterns in $\mathcal{P}_s \cup \mathcal{P}_+$ (i.e., all detectors of pattern $\boldsymbol{p}_i$):[5]

$$\mathcal{W}^+(i) = \left\{ j \mid \arg\max_{l \in \mathcal{P}_s \cup \mathcal{P}_+} \boldsymbol{w}^{(j)} \cdot \boldsymbol{p}_l = i, \ \boldsymbol{w}^{(j)} \cdot \boldsymbol{p}_i > 0 \right\}$$

$$\mathcal{U}^+(i) = \left\{ j \mid \arg\max_{l \in \mathcal{P}_s \cup \mathcal{P}_+} \boldsymbol{u}^{(j)} \cdot \boldsymbol{p}_l = i, \ \boldsymbol{u}_t^{(j)} \cdot \boldsymbol{p}_i > 0 \right\} \tag{2}$$

Next we define $M_{\boldsymbol{w}}^+(i) = \sum_{j \in \mathcal{W}^+(i)} \boldsymbol{w}^{(j)} \cdot \boldsymbol{p}_i$ and $M_{\boldsymbol{u}}^+(i) = \sum_{j \in \mathcal{U}^+(i)} \boldsymbol{u}^{(j)} \cdot \boldsymbol{p}_i$. The quantity $M_{\boldsymbol{w}}^+(i)$ is the sum of the dot products between a pattern $\boldsymbol{p}_i$ and its detectors $\boldsymbol{w}^{(j)}$. $M_{\boldsymbol{w}}^+(i)$ can be interpreted

---

[2] The order of the patterns will not matter, because the convolutional network is invariant to it.

[3] The reason we consider these two sets of patterns will be clear when we discuss detection ratios.

[4] Note that this does not affect the expressive power of the network.

[5] Given a tie between sets, we assume the weight is assigned arbitrarily to one of them.

as the overall response of $\boldsymbol{w}^{(j)}$ detectors of pattern $\boldsymbol{p}_i$. Similarly, we define $M_{\boldsymbol{u}}^+(i)$, which is like $M_{\boldsymbol{w}}^+(i)$, only with detectors $\boldsymbol{u}^{(j)}$.

For all $\boldsymbol{p}_i \in \mathcal{P}_+ \cup \mathcal{P}_s$ we refer to $\frac{M_{\boldsymbol{u}}^+(i)}{M_{\boldsymbol{w}}^+(1)}$ as *positive detection ratios*. The detection ratio can be interpreted as the ratio between the *undesired* response of pattern detectors of $\boldsymbol{p}_i$ and the *desired* response of discriminative pattern detectors (detectors of $\boldsymbol{p}_1$). Therefore, we would like this ratio to be small. Indeed, for any positive point $\boldsymbol{x}^+$ we have that:

$$N_W\left(\boldsymbol{x}^+\right) \geq M_{\boldsymbol{w}}^+(1) - \sum_{\boldsymbol{p}_i \in \mathcal{P}_s \cup \mathcal{P}_+} M_{\boldsymbol{u}}^+(i) = M_{\boldsymbol{w}}^+(1)\left(1 - \sum_{\boldsymbol{p}_i \in \mathcal{P}_s \cup \mathcal{P}_+} \frac{M_{\boldsymbol{u}}^+(i)}{M_{\boldsymbol{w}}^+(1)}\right) \tag{3}$$

where the inequality follows since positive points have only patterns in $\mathcal{P}_+ \cup \mathcal{P}_s$, by Eq. 1 and the definitions of $M_{\boldsymbol{w}}^+(i)$ and $M_{\boldsymbol{u}}^+(i)$. Notice that if all positive detection ratios are small, then the positive point is classified correctly. We will empirically show that for SGD, the magnitude of the detection ratios are governed by the statistics of the patterns in the training set, which will imply our generalization result.

Similarly, we define $\mathcal{W}^-(i), \mathcal{U}^-(i)$ and $M_{\boldsymbol{w}}^-(i), M_{\boldsymbol{u}}^-(i)$, where the only difference is using $\mathcal{P}_-$ instead of $\mathcal{P}_+$. Furthermore, for all $\boldsymbol{p}_i \in \mathcal{P}_- \cup \mathcal{P}_s$ we say that $\frac{M_{\boldsymbol{w}}^-(i)}{M_{\boldsymbol{u}}^-(2)}$ are *negative detection ratios*. Then, as in Eq. 3 we have for all negative points $\boldsymbol{x}^-$ that: $-N_W\left(\boldsymbol{x}^-\right) \geq M_{\boldsymbol{u}}^-(2)\left(1 - \sum_{\boldsymbol{p}_i \in \mathcal{P}_s \cup \mathcal{P}_-} \frac{M_{\boldsymbol{w}}^-(i)}{M_{\boldsymbol{u}}^-(2)}\right)$. This shows that if negative detection ratios are small, then all negative points are classified correctly. In the rest of the paper, we refer to both positive and negative detection ratios as detection ratios.

**Empirical Pattern Bias:** In a given training set, patterns will appear in both positive and negative examples. The following measure captures how well-balanced are the patterns between the labels. For any pattern $\boldsymbol{p}_i \in \mathcal{P}$, define the following statistic of the training set:

$$s_i = \frac{1}{m}\sum_{j=1}^m y_j \mathbb{1}\{\boldsymbol{p}_i \in \boldsymbol{x}_j\} \tag{4}$$

The detection ratios define quantities of the learned model. On the other hand, Eq. 4 is a quantity of the sampled training set. In the next section, we define the PSI property, which specifies how these two measures should be related to guarantee good generalization for the learned model.

## 4 Pattern Statistics Inductive Bias

The inductive bias of a learning algorithm refers to how the algorithm chooses among all models that fit the data equally well. For example, an SVM algorithm has an inductive bias towards low norm. Understanding the success of deep learning requires understanding the inductive bias of learning algorithms used to learn networks, and in particular SGD (Zhang et al., 2017).

In what follows, we define a certain inductive bias of an algorithm in our setting, which we refer to as the Patterns Statistics Inductive Bias (PSI) property. The PSI property states a simple relation between the relative frequency of patterns $s_i$ (see Eq. 4) and the detection ratios. We begin by providing the formal definition of PSI, and then provide further intuition. For the definition, we let $\mathcal{A}$ be any learning algorithm which given a training set $S$ returns a network $\mathcal{A}(S)$ as in Eq. 1.

**Definition 4.1.** *We say that a learning algorithm $\mathcal{A}$ satisfies the Patterns Statistics Inductive Bias condition with constants $b, c, \delta > 0$ ($(b,c,\delta)$-PSI) if the following holds. For any $m \geq 1$,* [6] *with probability at least $1 - \delta$ over the randomization of $\mathcal{A}$ and training set $S$ of size $m$, $\mathcal{A}(S)$ satisfies the following conditions:*

$$\forall i \in \mathcal{P}_s \cup \mathcal{P}_+ : \quad \frac{M_{\boldsymbol{u}}^+(i)}{M_{\boldsymbol{w}}^+(1)} \leq b \max\left(-\frac{s_i}{s_1}, 0\right) + \frac{c}{\sqrt{m}} \tag{5}$$

$$\forall i \in \mathcal{P}_s \cup \mathcal{P}_- : \quad \frac{M_{\boldsymbol{w}}^-(i)}{M_{\boldsymbol{u}}^-(2)} \leq b \max\left(-\frac{s_i}{s_2}, 0\right) + \frac{c}{\sqrt{m}} \tag{6}$$

---

[6] We state $m \geq 1$ for simplicity. Alternatively, one can assume $m \geq C$ for a constant $C$.

We next provide some *informal* intuition as to why SGD updates may lead to PSI (in Sec. 7 we provide a proof of this for a restricted setting).

We will consider updates made by gradient descent (full batch SGD). Define $\mathcal{W}_t^+(i)$ to be the set $\mathcal{W}^+(i)$ with weight vectors $\boldsymbol{w}_t^{(j)}$ instead of $\boldsymbol{w}^{(j)}$. Similarly, define $\mathcal{U}_t^+(i)$, $\mathcal{W}_t^-(i)$ and $\mathcal{U}_t^-(i)$. Throughout the discussion below, we assume that these sets have roughly the same size.[7] We will show that in certain cases, a high value of $-\frac{s_i}{s_1}$ implies that the detection ratio $\frac{M_{\boldsymbol{u}}^+(i)}{M_{\boldsymbol{w}}^+(1)}$ has a high value. Furthermore, a low value of $-\frac{s_i}{s_1}$ implies a low value of $\frac{M_{\boldsymbol{u}}^+(i)}{M_{\boldsymbol{w}}^+(1)}$. This motivates the bound in the PSI definition. As we will show, this follows since the statistics of the patterns in the training set $s_i$, govern the magnitude of the dot-product between a detector and its detected pattern.

By our distribution assumption we should have $s_1 \approx \frac{1}{2}$. First assume that $s_i \approx -\frac{1}{2}$ for $\boldsymbol{p}_i \in \mathcal{P}_s$, i.e., $-\frac{s_1}{s_i} \approx 1$. Now lets see what the detection ratio $\frac{M_{\boldsymbol{u}}^+(i)}{M_{\boldsymbol{w}}^+(1)}$ should be by the gradient update. Note that the gradient is a sum of updates, one for each point in the training set. Assume that $j \in \mathcal{W}_t^+(1)$, i.e., $\boldsymbol{w}^{(j)}$ detects $\boldsymbol{p}_1$. Then by the gradient update, the value $\frac{\eta}{m}\boldsymbol{p}_1$ is added to $\boldsymbol{w}_t^{(j)}$ for *all* positive points that have non-zero hinge loss. The value $-\frac{\eta}{m}\boldsymbol{p}_i$ is also added for a few $\boldsymbol{p}_i \in \mathcal{P}_- \cup \mathcal{P}_s$ and a subset of the negative points ($i$ depends on the specific negative point). In the next iteration, it holds that $j \in \mathcal{W}_{t+1}^+(1)$ and the updates continue similarly. Overall, we see that $\boldsymbol{w}_t^{(j)} \cdot \boldsymbol{p}_1$, which is the dot-product between the detector and its detected pattern, increases in each iteration and should be large after a few iterations. Therefore, $M_{\boldsymbol{w}}^+(1)$ should be large. By exactly the same argument, we should expect that for $j \in \mathcal{U}_t^+(i)$, $\boldsymbol{u}_t^{(j)} \cdot \boldsymbol{p}_i$ increases in each iteration and now $M_{\boldsymbol{u}}^+(i)$ should be large. Under the assumption that $|\mathcal{U}_t^+(i)| \approx |\mathcal{W}_t^+(1)|$, we should have $\frac{M_{\boldsymbol{u}}^+(i)}{M_{\boldsymbol{w}}^+(1)} \approx 1$ as well. Therefore, if $-\frac{s_1}{s_i} \approx 1$ then we should expect that $\frac{M_{\boldsymbol{u}}^+(i)}{M_{\boldsymbol{w}}^+(1)} \approx 1$.

On the other hand, if $\boldsymbol{p}_i$ appears in roughly an equal number of positive and negative points, i.e., $s_i \approx 0$, then we should expect $M_{\boldsymbol{u}}^+(i)$ to be low. To see this, consider a filter $j \in \mathcal{U}_t^+(i)$. In this case, positive points that contain $\boldsymbol{p}_i$ and with non-zero loss add $-\frac{\eta}{m}\boldsymbol{p}_i$ to $\boldsymbol{u}_t^{(j)}$, while negative points that contain $\boldsymbol{p}_i$ and have non-zero loss add $\frac{\eta}{m}\boldsymbol{p}_i$. Thus, $\boldsymbol{u}_t^{(j)} \cdot \boldsymbol{p}_i$ should not increase signficantly. Therefore, both $\frac{M_{\boldsymbol{u}}^+(i)}{M_{\boldsymbol{w}}^+(1)}$ and $\frac{-s_i}{s_1}$ should be small in this case.

Given the intuition above, one possible conjecture is that the detection ratio $\frac{M_{\boldsymbol{u}}^+(i)}{M_{\boldsymbol{w}}^+(1)}$ is bounded by an affine function of $\max\left(-\frac{s_i}{s_1}, 0\right)$, which leads to the PSI condition in Definition 4.1.[8] The bias term in the affine function takes into account that our intuition above is not exact. Finally, we can make a similar argument for $-\frac{s_i}{s_2}$ for motivating Eq. 6.

## 5 VC DIMENSION BOUND AND RELATION TO PSI

Here we show that the architecture in Section 3 is highly expressive, and can thus potentially overfit and generalize poorly. Moreover, we show examples of networks that overfit and do not satisfy PSI. First, a simple argument shows that $VC(\mathcal{H}) \leq \binom{d}{n}$ in our setting. The proof is given in Section A.

The lower bound below is more challenging, and reveals interesting connections to the PSI property.

**Theorem 5.1.** *Assume that $d = 2n$ and $n \geq 2$, then $VC(\mathcal{H}) \geq 2^{\frac{d}{2}-1}$.*

The full proof is given in Section B. Here we give a sketch. We construct a set $B$ of size $2^{n-1} = 2^{\frac{d}{2}-1}$ that can be shattered. For a given $I \in \{0,1\}^{n-1}$ let $I[j]$ be its $j$th entry. For any such $I$, define a point $\boldsymbol{x}_I$ such that for any $1 \leq j \leq n-1$, $\boldsymbol{x}_I[j] = I[j]\boldsymbol{p}_{2j+1} + (1 - I[j])\boldsymbol{p}_{2j+2}$. Furthermore, arbitrarily choose $\boldsymbol{x}_I[n] = \boldsymbol{p}_1$ or $\boldsymbol{x}_I[n] = \boldsymbol{p}_2$ and define $B = \{\boldsymbol{x}_I \mid I \in \{0,1\}^{n-1}\}$. Assume that each point $\boldsymbol{x}_I \in B$ has label $y_I$. We define a network with filters $\boldsymbol{w}^{(I)} = \max\{\alpha_I, 0\}\sum_{1 \leq j \leq n-1}\boldsymbol{x}_I[j]$ and

---

[7]This holds with high probability at initialization for a sufficiently large network. Furthermore, in Section 7, we show that it holds during training in the case of two training points.

[8]We consider $\max\left(-\frac{s_i}{s_1}, 0\right)$ in the PSI definition because the detection ratios are non-negative.

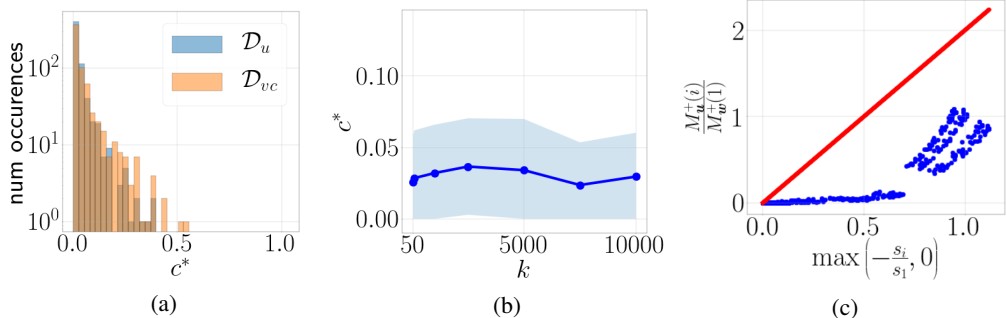

Figure 1: Empirical analysis of $c^*$. (a) Empirical calculation of $c^*$ for $\mathcal{D}_u$ and $\mathcal{D}_{vc}$. Values are in log scale. (b) $c^*$ as a function of the network size $k$ (c) Positive correlation between $\frac{M_u^+(i)}{M_w^+(1)}$ and $\max\left(-\frac{s_i}{s_1}, 0\right)$. The depicted line is the best PSI bound with $b = 2$ (lowest $c$).

$\boldsymbol{u}^{(I)} = \max\{-\alpha_I, 0\} \sum_{1 \le j \le n-1} \boldsymbol{x}_I[j]$ for each $I \in \{0,1\}^{n-1}$ and constants $\alpha_I$. Then, we prove that there exists constants $\alpha_I$ such that $N(\boldsymbol{x}_I) = y_I$ for all $I$ by solving a linear system.

**Relation to PSI:** Theorem 5.1 shows that there are exponentially large training sets that can be exactly fit with $\mathcal{H}$. This fact can be used to show a lower bound on sample complexity that is exponential in $d$ for general ERM algorithms (Anthony & Bartlett, 2009). The networks that fit these datasets are those defined by $\boldsymbol{w}^{(I)}, \boldsymbol{u}^{(I)}$. It is easy to see that these networks *do not* satisfy the PSI property. To see this, note that $M_w^+(1) = M_u^-(2) = 0$, which implies that the left-hand sides of parts 1 and 2 in the Definition 4.1 are infinite. Therefore, PSI is not satisfied for these networks.

These networks classify points based on the spurious patterns $\mathcal{P}_s$, and not on the patterns which determine the class. Networks that satisfy PSI are essentially the opposite: they classify a point mostly based on detectors for the patterns $\boldsymbol{p}_1$ and $\boldsymbol{p}_2$ and thus generalize well, as we show next.

## 6 PSI IMPLIES GOOD GENERALIZATION

In the previous section we showed that a general ERM algorithm for the class $\mathcal{H}$ may need exponentially many training samples to get low test error. Here we show that any algorithm satisfying the PSI condition (see Definition 4.1) will have low-degree polynomial sample complexity, when patterns in $\mathcal{P}_s$ are unbiased (i.e., $\mathbb{E}\left[y\mathbb{1}\{\boldsymbol{p}_i \in \boldsymbol{x}\}\right] = 0$ for $\boldsymbol{p}_i \in \mathcal{P}_s$). Specifically, in the following theorem we show that such an algorithm will have zero *test* error w.h.p., given only $O(|\mathcal{P}|^2 \log(|\mathcal{P}|))$ training samples. Note that this also implies a sample complexity of $O(d^2 \log(d))$ since $|\mathcal{P}| \le d$.

**Theorem 6.1.** *Assume that $\mathcal{D}$ satisfies the conditions in Section 3 and $\mathbb{E}\left[y\mathbb{1}\{\boldsymbol{p}_i \in \boldsymbol{x}\}\right] = 0$ for all $\boldsymbol{p}_i \in \mathcal{P}_s$. Let $\mathcal{A}$ be a learning algorithm which satisfies $(b,c,\delta)$-PSI with $b, c \ge 1$. Then, if $m > 300b^2c^2 |\mathcal{P}|^2 \log(|\mathcal{P}|)$, with probability at least $1 - \delta - \frac{4}{|\mathcal{P}|^3}$,[9] $\mathcal{A}(S)$ has $0$ test error.*

We defer the proof to the supplementary but here we sketch the main argument. By the assumption $\mathbb{E}\left[y\mathbb{1}\{\boldsymbol{p}_i \in \boldsymbol{x}\}\right] = 0$ and standard concentration of measure, $|s_i|$ should be small and therefore the detection ratios should be small by PSI. Then, by the key observation that small detection ratios imply perfect classification (e.g., Eq. 3), the algorithm achieves zero test error with respect to $\mathcal{D}$.

## 7 EMPIRICAL AND THEORETICAL EVIDENCE THAT SGD SATISFIES PSI

**Empirical Analysis:** Thus far we have established that the PSI property implies good generalization. Here we provide empirical evidence that SGD indeed learns such models with overparameterized CNNs. We also provide a qualitative analysis that further confirms that the statistics of the

---

[9]We note that the $\frac{4}{|\mathcal{P}|^3}$ may be improved to an arbitrary $\gamma > 0$ if we scale $m$ by $\log \frac{1}{\gamma}$.

patterns in the training set correlate with the detection ratios. Full details of the experiments are provided in the supplementary.

We perform experiments with two distributions denoted by $\mathcal{D}_u$ and $\mathcal{D}_{vc}$ that satisfy the properties defined in Section 3 and such that $\mathbb{E}\left[y\mathbb{1}\{\boldsymbol{p}_i \in \boldsymbol{x}\}\right] = 0$ for all $\boldsymbol{p}_i \in \mathcal{P}_s$. Thus, given Theorem 6.1, if PSI holds, good generalization will be implied. See Section E.2 for details on the distributions.

Next, we show that PSI holds with small constants $b$ and $c$ which do not change the order of magnitude of the bound in Theorem 6.1, i.e., $b^2 c^2 < 10$.[10] We trained a neural network in our setting with SGD as described in Section 3. We performed more than 1000 experiments with different parameter values for $n$, $d$, $k$ and $m$ (see Section E.3 for details) and performed 10 experiments for each set of values for $n$, $d$, $k$ and $m$. For each experiment, we set $b = 2$ and empirically calculated the lowest constant $c$ which satisfies the PSI definition, which we denote by $c^*$. The formal definition of $c^*$ is given in Eq. 10 in the supplementary. Figure 1a shows that across all experiments, the value of $c^*$ is less than 1, i.e., $b^2 (c^*)^2 < 10$. We further checked how $c^*$ varies with $k$ for $\mathcal{D} = \mathcal{D}_u$, $d = 50$ and $n = 20$. Figure 1b shows that $c^*$ is at most slightly correlated with $k$ and has low value for large $k$.

The intuition we described in Section 4 suggests that there is a positive correlation between $\frac{M_{\boldsymbol{u}}^+(i)}{M_{\boldsymbol{w}}^+(1)}$ and $\max\left(-\frac{s_i}{s_1}, 0\right)$. To test this, we experimented with a distribution which can vary the probability of a spurious pattern to be selected and thus can control $\max\left(-\frac{s_i}{s_1}, 0\right)$. Figure 1c clearly shows a positive correlation between these quantities, strongly suggesting that the statistics of the patterns in the training set govern the magnitude of the detection ratios. See Section E.5 for further details.

**Theoretical Analysis in a Simplified Setup:** Here we show that PSI holds for a setup of two training points, $S = \{(\boldsymbol{x}^+, 1), (\boldsymbol{x}^-, -1)\}$. We further assume that $\boldsymbol{x}^+$ and $\boldsymbol{x}^-$ have exactly the same patterns in $\mathcal{P}_s$. We analyze gradient descent with a constant learning rate $\eta = \frac{c_\eta}{k}$. The following theorem shows that PSI holds with constants $b = 1$ and $c = \sqrt{18}c_\eta$. The proof analyzes the trajectory of gradient descent and is provided in Section F.

**Theorem 7.1.** *For a sufficiently small $\epsilon$, $\sigma_g$, $c_\eta$ such that $\sigma_g << \eta$ and $k \geq poly\left(\log d, \frac{1}{\epsilon}\right)$, with probability at least $1 - \frac{9}{d^7} - 8e^{-8}$, gradient descent converges to a global minimum after $T \leq O\left(\frac{1}{c_\eta}\right)$ iterations and the PSI condition is satisfied with $b = 1$ and $c = \sqrt{18}c_\eta$.*

The theorem holds for overparameterized networks, which coincides with our empirical findings in Section 7. The theorem holds for sufficiently small initialization, and thus it is not in the same regime of NTK analysis where initialization is large (Woodworth et al., 2019; Chizat et al., 2019).

## 8 Experiment on MNIST

In this section we report experiments on a variant of MNIST (LeCun, 1998) and show that we can use our analysis to better understand the performance of overparameterized CNNs in this setting. Full details of the experiments are given in Section G.

Our PSI results thus far can be summarized informally as follows. In a pattern detection problem, an algorithm has PSI bias if the dot product between a discriminative pattern and its detector is large, and the dot product between a spurious pattern and its detector is low. Furthermore, the gap between these dot products increases with the train size and a sufficiently large gap implies perfect accuracy. While our analysis required the patterns to be orthogonal, the above idea can work beyond the orthogonal case, as the experiment below shows.

We consider data generated as follows. Each data point consists of 9 randomly sampled MNIST images, where if $y = 1$ one of the 9 digits is randomly chosen to be of color blue and the rest 8

---

[10]To empirically validate PSI and show that it implies good generalization, we could in principle show that the conditions of Theorem 6.1 hold empirically, i.e., there exist $b$, $c$ and $m$ such that $m > 300b^2c^2d^2\log(d)$ and PSI holds with constants $b$, $c$ and high probability $1 - \delta$. However, as with most generalization results, the numerical value (including constants) results in large $m$ which cannot be empirically tested. Instead, we show that $b$ and $c$ do not change the order of magnitude of the bound.

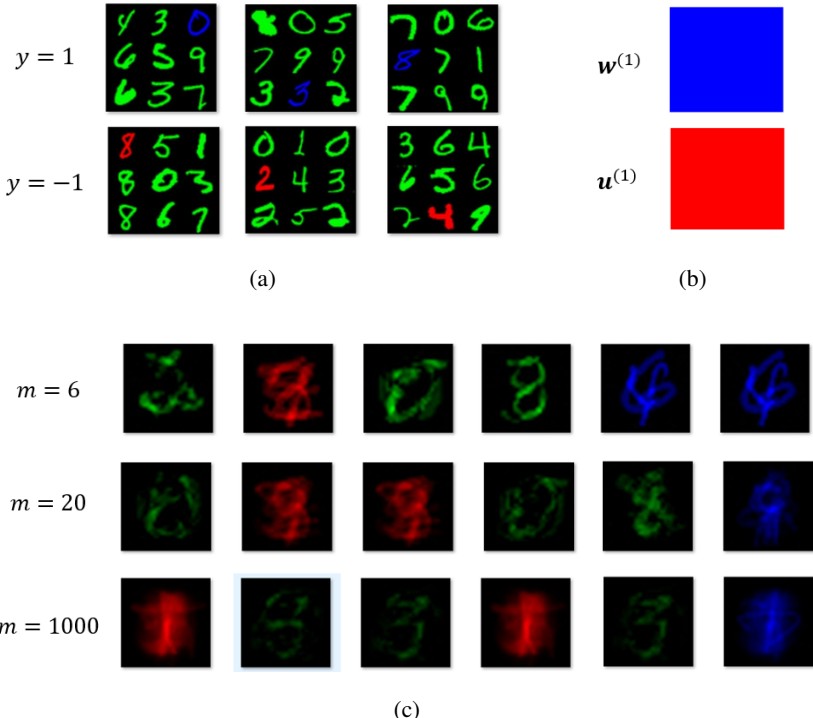

Figure 2: Experiments on a variant of MNIST. (a) Examples of data points. (b) Filters of CNN with $k = 1$ that perfectly classifies the data (c) Examples of learned filters of overparameterized CNNs for different training set sizes. Figures of all learned filters are given in Section G.

digits are colored green. For $y = -1$ a similar sampling procedure is performed but with the color red instead of blue. Figure 2a shows examples of data points (note that in our notation $n = 9$ and $d = 28 * 28 * 3 = 2352$). Thus in this setting, red and blue digits are discriminative whereas green are spurious.

We use the network in Eq. 1 and SGD to learn a classifier for this data. The data can be perfectly classified with a network with $k = 1$ (see Figure 2b). We train an overparameterized network with $k = 20$ for different training set sizes. Figure 2c shows subsets of the learned filters (see Section G for all filters), for different training set sizes. The figures show in color the positive weight filter entries. The pattern that appears is the pattern that maximally activates them, i.e., the pattern they detect.

First, we can see that the filters come in three colors (blue, red, green) corresponding to the three pattern types they detect (positive, negative, spurious respectively). This fact is not trivial and is similar to what we obtained in the proof of Theorem 7.1 and explained in Section 4.

As noted above, the PSI prediction is that the dot product between detectors and detected patterns would be large for discriminative patterns (i.e., red and blue) and low for spurious (i.e., green). Furthermore, this difference should increase with the data size $m$. Indeed, Figure 2c shows precisely this behavior. Namely, as we increase the training set size, the green pattern detectors become darker and thus have low dot product with detected pattern. In contrast, the red and blue maintain a bright color and thus have large dot products with detected patterns. Finally, the test accuracy for $m = 6, 20, 1000$ is $88\%, 100\%, 100\%$, respectively. This is in line with Theorem 6.1 that shows PSI can lead to perfect test accuracy when the gap between dot products is sufficiently large.

## 9  CONCLUSIONS

Understanding the inductive bias of gradient methods for deep learning is an important challenge. In this paper, we study the inductive bias of overparameterized CNNs in a novel setup and provide

theoretical and empirical support that SGD exhibits good generalization performance. Our results on MNIST suggest that the PSI phenomenon goes beyond orthogonal patterns.

We use a unique approach of combining novel empirical observations with theoretical guarantees to make headway in a challenging setting of overparameterized CNNs. We believe that our work can pave the way for studying inductive bias of neural networks in other challenging settings.

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

## A  VC Dimension Upper Bound

Without considering the order of the patterns in the images, there are at most $\binom{d}{n}$ input points in $\mathcal{D}$. Since the network in Eq. 1 is invariant to the order of the patterns in an image, this implies: $VC(\mathcal{H}) \le \binom{d}{n}$. Note that for the definition of VC dimension, we assume that the domain of possible inputs is the domain of images of the distribution. This gives a tighter upper bound for our problem.

## B  Proof of Theorem 5.1

We will construct a set $B$ of size $2^{n-1} = 2^{\frac{d}{2}-1}$ that can be shattered. For a given $I \in \{0,1\}^{n-1}$ let $I[j]$ be its $j$th entry. For any such $I$, define a point $x_I$ such that for any $1 \le j \le n-1$, $x_I[j] = I[j]p_{2j+1} + (1-I[j])p_{2j+2}$. Furthermore, arbitrarily choose $x_I[n] = p_1$ or $x_I[n] = p_2$ and define $B = \{x_I \mid I \in \{0,1\}^{n-1}\}$.

Now, assume that each point $x_I \in B$ has label $y_I$. We will show that there is a network $N \in \mathcal{H}$ such that $N(x_I) = y_I$ for all $I$. For each $I \in \{0,1\}^{n-1}$, define $w^{(I)} = \max\{\alpha_I, 0\} \sum_{1 \le j \le n-1} x_I[j]$ and $u^{(I)} = \max\{-\alpha_I, 0\} \sum_{1 \le j \le n-1} x_I[j]$, where $\{\alpha_I\}$ is the unique solution of the following linear system with $2^{n-1}$ equations. For each $I \in \{0,1\}^{n-1}$ the system has the following equation:

$$\sum_{I' \in \{0,1\}^{n-1} \setminus \{I\}} \alpha_{I'} = y_{I^c} \tag{7}$$

where for any $I \in \{0,1\}^{n-1}$, $I^c \in \{0,1\}^{n-1}$ is defined such that $I^c[j] = 1 - I[j]$ for all $1 \le j \le n-1$. There is a unique solution because the corresponding matrix of the linear system is the difference between an all 1's matrix and the identity matrix. By the Sherman-Morrison formula (Sherman & Morrison, 1950), this matrix is invertible, where in the formula the outer product rank-1 matrix is the all 1's matrix and the invertible matrix is minus the identity matrix. Then for $N$ with the above weights and any $x_I$:

$$N(x_I) = \sum_{I' \in \{0,1\}^{n-1}} \left[ \max_j \left\{ \sigma\left(w^{(I')} \cdot x[j]\right) \right\} - \max_j \left\{ \sigma\left(u^{(I')} \cdot x[j]\right) \right\} \right]$$

$$= \sum_{I' \in \{0,1\}^{n-1}} \alpha_{I'} \max_j \left\{ \sigma\left(\sum_{1 \le i \le n-1} x_{I'}[i] \cdot x_I[j]\right) \right\} = \sum_{I' \in \{0,1\}^{n-1} \setminus \{I^c\}} \alpha_{I'} = y_I$$

by the definition of $N$, the orthogonality of the patterns, and Eq. 7. We have shown that any labeling $y_I$ can be achieved, and hence the set is shattered, completing the proof.

## C  Proof of Theorem 6.1

WLOG we prove the theorem for $|\mathcal{P}| = d$. By the assumption, for $p_i \in \mathcal{P}_s$, $s_i$ is an average of $m$ IID binary variables $y_j \mathbb{1}\{p_i \in x_j\}$ with zero expected value. Thus, by Hoeffding's inequality we have for all $p_i \in \mathcal{P}_s$ that:

$$\mathbb{P}\left( b|s_i| \le 4b\sqrt{\frac{\log(d)}{m}} \right) \le \frac{2}{d^4}$$

Therefore, by a union bound over all patterns $p_i \in \mathcal{P}_s$, with probability at least $1 - \frac{2}{d^3}$, for all $p_i \in \mathcal{P}_s$:

$$b|s_i| \le 4b\sqrt{\frac{\log(d)}{m}} \le \frac{1}{6cd} \tag{8}$$

Next we consider $p_1$ (the positive pattern), for which $\mathbb{E}[s_1] = 0.5$ (because it only appears in the positive examples, and the prior over $y$ is 0.5). Hoeffding's bound and the definition of $m$ imply that $|s_1| \ge \frac{1}{3}$ with probability at least $1 - \frac{1}{d^3}$.[11]

---

[11]In fact we can have exponential dependence here, but we use $d^3$ to simplify later expressions.

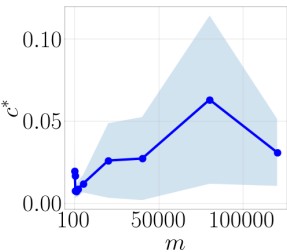

Figure 3: $c^*$ as a function of the training set size $m$.

We can now do a union bound over all patterns and PSI condition to obtain that with probability at least $1 - \delta - \frac{3}{d^3}$ we have by the PSI property and Eq. 8, for all $p_i \in \mathcal{P}_s$:

$$\frac{M_u^+(i)}{M_w^+(1)} \leq b \frac{|s_i|}{|s_1|} + \frac{c}{\sqrt{m}} \leq \frac{1}{2cd} + \frac{c}{\sqrt{m}} < \frac{1}{d} \tag{9}$$

From PSI we have $\frac{M_u^+(1)}{M_w^+(1)} \leq \frac{c}{\sqrt{m}} < \frac{1}{d}$. Therefore, for any positive point $(x^+, 1)$ Eq. 3 implies:

$$N_W(x^+) > M_w^+(1) \left(1 - \frac{d-1}{d}\right) > 0$$

Thus, $x^+$ is classified correctly. By the symmetry of the problem and part 2 in Definition 4.1, any negative point will be classified correctly as well.

## D  FURTHER EXPERIMENTS FOR VALIDATION OF PSI

To further validate the PSI condition, we tested whether the conditions in the proof of Theorem 6.1 empirically hold. Specifically, in the proof we showed that $\frac{M_u^+(i)}{M_w^+(1)} < \frac{1}{d}$ for all $p_i \in \mathcal{P}_s$ (in Eq. 9). We checked this for all settings of $(n, d)$ and largest possible $k$ and $m$, $k = 10000$ and $m = 40000$. In all of our experiments, SGD converged to a solution with 0 test error such that Eq. 9 holds for all $p_i \in \mathcal{P}_s$.

Finally, we checked how $c^*$ varies with $m$. Figure 3 show that $c^*$ is at most slightly correlated with $m$ and has low value for large $m$. In the same setup of Section E.3, we performed experiments with distribution $\mathcal{D}_u$, $n = 20$, $d = 50$, $k = 2500$ and $m \in \{100, 200, 500, 1000, 2000, 5000, 20000, 40000, 80000, 120000\}$.

## E  EXPERIMENTAL DETAILS OF SECTION 7

Here we provide details of the experiments performed in Section 7. All experiments were run on NVidia Titan Xp GPUs with 12GB of memory. Training algorithms were implemented in Tensor-Flow. All of the empirical results can be replicated in approximately 150 hours on a single Nvidia Titan Xp GPU.

### E.1  VALUE OF $c^*$

We use the following formula to compute $c^*$ in the experiments:

$$c^* = \sqrt{m} \max \left\{ \max_{i \in \mathcal{P}_s \cup \mathcal{P}_+} \frac{M_u^+(i)}{M_w^+(1)} - 2 \max \left(-\frac{s_i}{s_1}, 0\right), \max_{i \in \mathcal{P}_s \cup \mathcal{P}_-} \frac{M_w^-(i)}{M_u^-(2)} - 2 \max \left(-\frac{s_i}{s_2}, 0\right), 0 \right\} \tag{10}$$

### E.2  DISTRIBUTIONS IN EXPERIMENTS

We perform experiments with two types of distributions that satisfy the properties defined in Section 3. They differ in the random sampling procedure of spurious patterns described in Section 3. In both

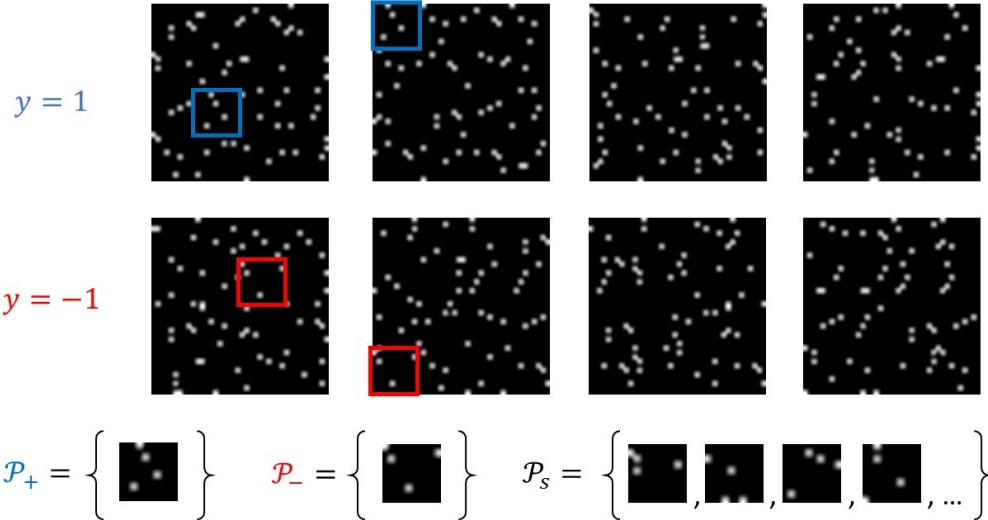

Figure 4: Example of points in an orthogonal patterns distribution. Here there are 25 possible orthogonal patterns ($|\mathcal{P}| = 25$) and each pattern is a $10 \times 10$ image patch and thus $d = 100$. The number of patterns in each image is $n = 16$. The image consists of 4 rows of 4 patches each. The positive examples contain the pattern in $\mathcal{P}_+$. The negative examples contain the pattern in $\mathcal{P}_-$. In the two leftmost images of each class, the corresponding pattern is shown. All other patterns in an image are from the set of spurious patterns $\mathcal{P}_s$.

distributions $\mathcal{P}$ is the set of all one-hot vectors in $\mathbb{R}^d$. In the first distribution $\mathcal{D}_u$, the $n - 1$ spurious patterns are selected uniformly at random without replacement from $\mathcal{P}_s$. In the second distribution $\mathcal{D}_{vc}$, for each $1 \leq j \leq n - 1$ one of the patterns from $\{\boldsymbol{p}_{2j+1}, \boldsymbol{p}_{2j+2}\}$ is selected uniformly at random. Importantly, both $\mathcal{D}_u$ and $\mathcal{D}_{vc}$ satisfy $\mathbb{E}\left[y\mathbb{1}\{\boldsymbol{p}_i \in \boldsymbol{x}\}\right] = 0$ for all $\boldsymbol{p}_i \in \mathcal{P}_s$. Thus, given Theorem 6.1, if PSI holds, good generalization will be implied.

**Remark E.1.** *The support of $\mathcal{D}_{vc}$ is the shattered set $B$ in the proof of Theorem 5.1. The proof implies that for any sampled training and test sets which are subsets of $B$, there exists a network with 0 training error and arbitrarily high test error. Therefore, by optimizing the training error, SGD can converge to these solutions. However, as we show empirically, SGD does not converge to these solutions, but rather it satisfies PSI and converges to solutions with good generalization performance.*

### E.3    FIGURE 1A EXPERIMENT

We performed more than 1000 experiments with the network in Eq. 1 and SGD. We experimented with parameter values $k \in \{1000, 10000\}$, $m \in \{100, 500, 1000, 2000, 5000, 20000, 40000\}$, $(n, d) \in \{(10, 20), (10, 80), (20, 50), (40, 60)\}$ for $\mathcal{D} = \mathcal{D}_u$ and $(n, d) \in \{(10, 20), (40, 80), (25, 50), (30, 60)\}$ for $\mathcal{D} = \mathcal{D}_{vc}$. For each distribution $\mathcal{D}_u$ or $\mathcal{D}_{vc}$, we performed 10 experiments for each set of values for $n$, $d$, $k$ and $m$. For each set of values we plot the mean of the 10 experiments and standard deviation error bars in shaded regions. In each one of the 10 experiments we randomly sampled the training and test sets according to the given distribution $\mathcal{D}_u$ or $\mathcal{D}_{vc}$ and randomly sampled the initialization of the network. We used a test set of size 1000. All orthogonal patterns were one-hot vectors. We trained only the weights of the first convolutional layer. We used a batch size of 20 if $k = 10000$ and batch size of 100 for $k = 1000$. The learning rate was set to $\max\{\frac{0.001}{2k}, 0.0000001\}$ and $\sigma_g$ to 0.000001. The solution SGD returned was either after 50000 epochs or if there was an epoch where the training loss was less than 0.00001. For each experiment, we set $b = 2$ and empirically calculated $c^*$.

### E.4 FIGURE 1B EXPERIMENT

In the same setup of Section E.3 (i.e., batch size, stopping criteria, learning rate etc.), we performed experiments with distribution $\mathcal{D}_u$, $n = 20$, $d = 50$, $m = 2000$ and $k \in \{50, 100, 1000, 2500, 5000, 7500, 10000\}$.

### E.5 FIGURE 1C EXPERIMENT

We experimented with a distribution $\mathcal{D}_p$ which can vary the probability of a spurious pattern to be selected and thus can control $\max\left(-\frac{s_i}{s_1}, 0\right)$. Given $y = 1$ it selects $\boldsymbol{p}_3$ with probability $p$ or $\boldsymbol{p}_4$ with probability $1 - p$. Then it selects the remaining $n - 2$ patterns from $\mathcal{P}_s \setminus \{\boldsymbol{p}_3, \boldsymbol{p}_4\}$ uniformly at random without replacement. Similarly, given $y = -1$ it selects $\boldsymbol{p}_3$ with probability $1 - p$ or $\boldsymbol{p}_4$ with probability $p$. The remaining $n - 2$ patterns are selected uniformly without replacement from $\mathcal{P}_s \setminus \{\boldsymbol{p}_3, \boldsymbol{p}_4\}$. We experimented with various $p$ and plotted for each solution of SGD, $\frac{M_{\boldsymbol{u}}^+(i)}{M_{\boldsymbol{w}}^+(1)}$ and $\max\left(-\frac{s_i}{s_1}, 0\right)$ for all $\boldsymbol{p}_i \in \mathcal{P}_s \cup \mathcal{P}_+$.

In the setup of Section E.3 we experimented with distributions $\mathcal{D}_p$ for $p$ values in

$$\{0.0, 0.01, 0.03, 0.05, 0.07, 0.1, 0.12, 0.2, 0.21, 0.28, 0.3, 0.4,$$
$$0.44, 0.5, 0.51, 0.59, 0.6, 0.68, 0.7, 0.78, 0.8, 0.9, 0.91, 0.94$$
$$0.95, 0.98, 0.99, 1.0\}$$

We experimented with values $n = 40$, $d = 60$, $m = 1000$ and $k = 2500$. The solution SGD returned was either after 2000 epochs or if there was an epoch where the training loss was less than $0.00001$.

## F PROOF OF THEOREM 7.1

### F.1 NOTATIONS

Here we define additional notations that will be useful for the proof of the theorem.

Let $\mathcal{P}_T$ be the set of all patterns that appear in either $x^+$ or $x^-$. Similarly to Eq. 2 define:

$$\mathcal{W}_t^+(i) = \left\{ j \mid \underset{l \in \mathcal{P}_T \setminus \{2\}}{\arg\max} \, \boldsymbol{w}_t^{(j)} \cdot \boldsymbol{p}_l = i, \, \boldsymbol{w}_t^{(j)} \cdot \boldsymbol{p}_i > 0 \right\}$$
$$\mathcal{U}_t^+(i) = \left\{ j \mid \underset{l \in \mathcal{P}_T \setminus \{2\}}{\arg\max} \, \boldsymbol{u}_t^{(j)} \cdot \boldsymbol{p}_l = i, \, \boldsymbol{u}_t^{(j)} \cdot \boldsymbol{p}_i > 0 \right\} \tag{11}$$

and

$$\mathcal{W}_t^-(i) = \left\{ j \mid \underset{l \in \mathcal{P}_T \setminus \{1\}}{\arg\max} \, \boldsymbol{w}_t^{(j)} \cdot \boldsymbol{p}_l = i, \, \boldsymbol{w}_t^{(j)} \cdot \boldsymbol{p}_i > 0 \right\}$$
$$\mathcal{U}_t^-(i) = \left\{ j \mid \underset{l \in \mathcal{P}_T \setminus \{1\}}{\arg\max} \, \boldsymbol{u}_t^{(j)} \cdot \boldsymbol{p}_l = i, \, \boldsymbol{u}_t^{(j)} \cdot \boldsymbol{p}_i > 0 \right\} \tag{12}$$

Define:

$$A_{\boldsymbol{w}} = \bigcup_{i \in \mathcal{P}_T \setminus \{2\}} \mathcal{W}_0^+(i)$$
$$A_{\boldsymbol{u}} = \bigcup_{i \in \mathcal{P}_T \setminus \{1\}} \mathcal{U}_0^-(i)$$

Finally we define $poly(x)$ to be any polynomial function of $x$.

### F.2 AUXILIARY LEMMAS

We now prove several technical lemmas. In Section F.3 we use the lemmas to prove the theorem. In the next 3 lemmas we provide high probability bounds on sizes of certain sets that are functions of the sets in Eq. 11 and Eq. 12.

**Lemma F.1.** *For any $0 < \epsilon < \frac{1}{4}$, with probability at least $1 - 4e^{-8}$ for any $k > poly(\frac{1}{\epsilon})$:*

$$\left| \frac{|A_{\boldsymbol{w}}|}{k} - (1 - 2^{-n}) \right| \leq \epsilon$$

*and*

$$\left| \frac{|A_{\boldsymbol{u}}|}{k} - (1 - 2^{-n}) \right| \leq \epsilon$$

*Proof.* It suffices to show that for any $k$:

$$\left| |A_{\boldsymbol{w}}| - (1 - 2^{-n}) \, k \right| \leq 2\sqrt{k}$$

and

$$\left| |A_{\boldsymbol{u}}| - (1 - 2^{-n}) \, k \right| \leq 2\sqrt{k}$$

For each $1 \leq j \leq k$ it holds that $j \in A_{\boldsymbol{w}}$ with probability $1 - 2^{-n}$. Therefore by Hoeffding's inequality, with probability at least $1 - 2e^{-8}$,

$$\left| |A_{\boldsymbol{w}}| - (1 - 2^{-n}) \, k \right| \leq 2\sqrt{k}.$$

The same argument applies for $|A_{\boldsymbol{u}}|$, a union bound and setting $k > \frac{1}{\epsilon^3}$ concludes the proof. $\qquad\square$

**Lemma F.2.** *For any $\epsilon > 0$, with probability at least $1 - \frac{4}{d^7} - 4e^{-8}$, for $k > poly(\log d, \frac{1}{\epsilon})$ and for all $i \in \mathcal{P}_T \setminus \{2\}$:*

$$\frac{|\mathcal{W}_0^+(i)|}{k(1 - 2^{-n} + \epsilon)} \leq \frac{1}{n} + \epsilon$$

*and*

$$\frac{|\mathcal{W}_0^+(i)|}{k(1 - 2^{-n} - \epsilon)} \geq \frac{1}{n} - \epsilon$$

*Similarly, for all $i \in \mathcal{P}_T \setminus \{1\}$:*

$$\frac{|\mathcal{U}_0^-(i)|}{k(1 - 2^{-n} + \epsilon)} \leq \frac{1}{n} + \epsilon$$

*and*

$$\frac{|\mathcal{U}_0^-(i)|}{k(1 - 2^{-n} - \epsilon)} \geq \frac{1}{n} - \epsilon$$

*Proof.* Without loss of generality, consider $|\mathcal{W}_0^+(i)|$. We first condition on the random variable $|A_{\boldsymbol{w}}|$ and given that the event $k(1 - 2^{-n} - \epsilon) \leq |A_{\boldsymbol{w}}| \leq k(1 - 2^{-n} + \epsilon)$ holds. By symmetry, we have $\mathbb{E}\left[\frac{|\mathcal{W}_0^+(i)|}{|A_{\boldsymbol{w}}|}\right] = \frac{1}{n}$ where the expectation is with respect to the initialization. Thus, we get by Hoeffding's inequality:

$$\mathbb{P}\left( \left| \frac{|\mathcal{W}_0^+(i)|}{|A_{\boldsymbol{w}}|} - \frac{1}{n} \right| \leq \frac{2\sqrt{\log d}}{\sqrt{|A_{\boldsymbol{w}}|}} \right)$$

$$\leq 2e^{-2|A_{\boldsymbol{w}}|\left(\frac{2\sqrt{\log d}}{\sqrt{|A_{\boldsymbol{w}}|}}\right)^2} = \frac{2}{d^8}$$

By the law of total probability, applying Lemma F.1 and a union bound over $i \in \mathcal{P}_T$ twice (for both $\mathcal{W}_0^+(i)$ and $\mathcal{U}_0^-(i)$), we get the desired result. $\qquad\square$

**Lemma F.3.** *For any $\epsilon > 0$, with probability at least $1 - \frac{4}{d^7} - 4e^{-8}$, for $k > poly(\log d, \frac{1}{\epsilon})$ and for all $i \in \mathcal{P}_T \setminus \{1, 2\}$ the following holds:*

$$\frac{|\mathcal{W}_0^+(i) \cap \mathcal{W}_0^-(2)|}{k\left(\frac{1}{2} - 2^{-n-1} + \epsilon\right)} \leq \frac{1}{n(n+1)} + \epsilon$$

$$\frac{|\mathcal{W}_0^+(i) \cap \mathcal{W}_0^-(2)|}{k\left(\frac{1}{2} - 2^{-n-1} - \epsilon\right)} \geq \frac{1}{n(n+1)} - \epsilon$$

$$\frac{|\mathcal{U}_0^+(1) \cap \mathcal{U}_0^-(i)|}{k\left(\frac{1}{2} - 2^{-n-1} + \epsilon\right)} \leq \frac{1}{n(n+1)} + \epsilon$$

*and*

$$\frac{|\mathcal{U}_0^+(1) \cap \mathcal{U}_0^-(i)|}{k\left(\frac{1}{2} - 2^{-n-1} - \epsilon\right)} \geq \frac{1}{n(n+1)} - \epsilon$$

*Proof.* The proof is similar to the proofs of Lemma F.1 and Lemma F.2. The difference is that we use the equalities $\mathbb{E}\left[|A_{\boldsymbol{w}} \cap \mathcal{W}_0^-(2)|\right] = \mathbb{E}\left[|A_{\boldsymbol{u}} \cap \mathcal{U}_0^+(1)|\right] = \left(\frac{1}{2} - 2^{-n-1}\right)k$ instead of $\mathbb{E}\left[|A_{\boldsymbol{w}}|\right] = \mathbb{E}\left[|A_{\boldsymbol{u}}|\right] = (1 - 2^{-n})k$ as in Lemma F.1. Furthermore, we use $\mathbb{E}\left[\frac{|\mathcal{W}_0^+(i) \cap \mathcal{W}_0^-(2)|}{|A_{\boldsymbol{w}} \cap \mathcal{W}_0^-(2)|}\right] = \mathbb{E}\left[\frac{|\mathcal{U}_0^+(1) \cap \mathcal{U}_0^-(i)|}{|A_{\boldsymbol{u}} \cap \mathcal{U}_0^+(1)|}\right] = \frac{1}{n(n+1)}$ for fixed $|A_{\boldsymbol{w}} \cap \mathcal{W}_0^-(2)|$ and $|A_{\boldsymbol{u}} \cap \mathcal{U}_0^+(1)|$ instead of $\mathbb{E}\left[\frac{|\mathcal{W}_0^+(i)|}{|A_{\boldsymbol{w}}|}\right] = \mathbb{E}\left[\frac{|\mathcal{U}_0^-(i)|}{|A_{\boldsymbol{u}}|}\right] = \frac{1}{n}$ for fixed $|A_{\boldsymbol{w}}|$ and $|A_{\boldsymbol{u}}|$ as in Lemma F.2. $\qquad\square$

**Lemma F.4.** *For any $M > 0$ and $\delta > 0$, there exists a sufficiently small $\sigma_g > 0$, such that with probability at least $1 - \delta$, for all $1 \leq i \leq k$, $\left\|\boldsymbol{w}_0^{(i)}\right\| \leq M$ and $\left\|\boldsymbol{u}_0^{(i)}\right\| \leq M$.*

*Proof.* The proof is immediate. $\qquad\square$

We now proceed to analyze the dynamics of gradient descent in the next two lemmas. Define $M$ such that for all $1 \leq i \leq k$, $\left\|\boldsymbol{w}_0^{(i)}\right\| \leq M$ and $\left\|\boldsymbol{u}_0^{(i)}\right\| \leq M$. Let $\mathcal{E}$ be the set of all $t$ such that for all $\boldsymbol{x} \in S$, it holds that $N_{W_t}(\boldsymbol{x}) < 1$. Let $t^* = \arg\min_t \{t - 1 \in \mathcal{E}, t \notin \mathcal{E}\}$. We assume that $\eta$ and $\sigma_g$ are sufficiently small such that $t^* \geq 2$.

**Lemma F.5.** *For a sufficiently small $\epsilon$, $M$, $c_\eta$ such that $M \ll \eta$, the following holds for any $1 \leq t \leq t^*$:*

1. *If $j \notin A_{\boldsymbol{w}}$, then $\boldsymbol{w}_t^{(j)} = \boldsymbol{w}_0^{(j)} - \alpha \frac{\eta}{2} \boldsymbol{p}_2$ where $\alpha \in \{0, 1\}$.*

2. *If $j \in \mathcal{W}_0^+(1)$, then $\boldsymbol{w}_t^{(j)} = \boldsymbol{w}_0^{(j)} - \frac{\eta}{2} \sum_{i \in \mathcal{P}_T \setminus \{1\}} \alpha_i \boldsymbol{p}_i + \frac{\eta t}{2} \boldsymbol{p}_1$, where $\alpha_i \in \{0, 1\}$.*

3. *If $i \in \mathcal{P}_T \cap \mathcal{P}_s$ and $j \in \mathcal{W}_0^+(i) \cap \mathcal{W}_0^-(i)$ then $\boldsymbol{w}_t^{(j)} = \boldsymbol{w}_0^{(j)}$.*

4. *If $i \in \mathcal{P}_T \cap \mathcal{P}_s$ and $j \in \mathcal{W}_0^+(i) \cap \mathcal{W}_0^-(2)$, then $\boldsymbol{w}_t^{(j)} = \boldsymbol{w}_0^{(j)} - \frac{\eta}{2} \boldsymbol{p}_2 + \frac{\eta}{2} \boldsymbol{p}_i$*

5. *If $j \notin A_{\boldsymbol{u}}$, then $\boldsymbol{u}_t^{(j)} = \boldsymbol{u}_0^{(j)} - \alpha \boldsymbol{p}_1$ where $\alpha \in \{0, 1\}$.*

6. *If $j \in \mathcal{U}_0^-(2)$, then $\boldsymbol{u}_t^{(j)} = \boldsymbol{u}_0^{(j)} - \frac{\eta}{2} \sum_{i \in \mathcal{P}_T \setminus \{2\}} \alpha_i \boldsymbol{p}_i + \frac{\eta t}{2} \boldsymbol{p}_2$, where $\alpha_i \in \{0, 1\}$.*

7. *If $i \in \mathcal{P}_T \cap \mathcal{P}_s$ and $j \in \mathcal{U}_0^-(i) \cap \mathcal{U}_0^+(1)$ then $\boldsymbol{u}_t^{(j)} = \boldsymbol{u}_0^{(j)}$.*

8. *If $i \in \mathcal{P}_T \cap \mathcal{P}_s$ and $j \in \mathcal{U}_0^-(i) \cap \mathcal{U}_0^+(1)$, then $\boldsymbol{u}_t^{(j)} = \boldsymbol{u}_0^{(j)} - \frac{\eta}{2} \boldsymbol{p}_1 + \frac{\eta}{2} \boldsymbol{p}_i$*

*Proof.* 1. If $j \notin \mathcal{W}_0^-(2)$, then for $t = 1$ the gradient of the loss with respect to $\boldsymbol{w}^{(i)}$ is 0, because every pattern in $\mathcal{P}_T$ has a negative dot product with $\boldsymbol{w}_0^{(i)}$. Therefore, $\boldsymbol{w}_1^{(i)} = \boldsymbol{w}_0^{(i)}$. By the same argument $\boldsymbol{w}_t^{(i)} = \boldsymbol{w}_0^{(i)}$ for all $t \geq 1$. If $j \in \mathcal{W}_0^-(2)$ then $\frac{\eta}{2} \boldsymbol{p}_2$ will subtracted in the first iteration, and $\boldsymbol{w}_t^{(i)}$ will not change in later iterations.

2. The proof follows directly by the gradient update. In each iteration, $\frac{\eta}{2}\boldsymbol{p}_1$ is added and a pattern in $\mathcal{P}_T \smallsetminus \{1\}$ is subtracted unless all such patterns already have a negative dot product with $\boldsymbol{w}_t^{(i)}$. Note that we used here the fact that $M << \eta$.

3. For $t = 1$ we have by the gradient update:

$$\boldsymbol{w}_1^{(j)} = \boldsymbol{w}_0^{(j)} + \frac{\eta}{2}\boldsymbol{p}_i - \frac{\eta}{2}\boldsymbol{p}_i = \boldsymbol{w}_0^{(j)} \tag{13}$$

By induction, $\boldsymbol{w}_t^{(i)} = \boldsymbol{w}_0^{(i)}$ for all $1 \le t \le t^*$.

4. The proof follows by the gradient update as in previous proofs. For $t = 1$, the term $\frac{\eta}{2}\boldsymbol{p}_2$ is subtracted by the update of $\boldsymbol{x}^-$, since $j \in \mathcal{W}_0^-(2)$. The term $\frac{\eta}{2}\boldsymbol{p}_i$ is added due to the update of $\boldsymbol{x}^+$. Now $j \in \mathcal{W}_1^+(i) \cap \mathcal{W}_1^-(i)$ and thus $\boldsymbol{w}_t^{(j)}$ will not change in subsequent iterations, as in the proof of part 3. This concludes the proof.

By symmetry, the proofs of 5-8 are identical to the proofs of parts 1-4. $\qquad\square$

Define

$$\gamma = \frac{(1 - 2^{-n})}{n} \tag{14}$$

then we have the following:

**Lemma F.6.** *For a sufficiently small $\epsilon$, $M$, $c_\eta$ such that $M << \eta$ and $k \ge poly\left(\log d, \frac{1}{\epsilon}\right)$ with probability at least $1 - \frac{9}{d^7} - 8e^{-8}$, gradient descent converges to a global minimum after $\frac{1}{\gamma c_\eta} \le T \le \frac{3}{\gamma c_\eta}$ iterations.*

*Proof.* Throughout the proof we use Lemma F.4 to choose a sufficiently small $M$ such that $\left|\boldsymbol{w}_0^{(i)}\right| \le M$ and $\left|\boldsymbol{u}_0^{(i)}\right| \le M$ with probablity at least $1 - \frac{1}{d^7}$. We further apply Lemma F.1, Lemma F.2 and Lemma F.3 which together with Lemma F.4 hold with probability at least $1 - \frac{9}{d^7} - 8e^{-8}$. Define the sets of weights $B_i$ such that $i$ corresponds to the set of weights in part $i$ of Lemma F.5. For example, $B_2 = \mathcal{W}_0^+(1)$ and $B_8 = \bigcup_{i \in \mathcal{P}_T \cap \mathcal{P}_s} \mathcal{U}_0^-(i) \cap \mathcal{U}_0^+(1)$.

Define the following:

$$N_{W_t}^{(1)}(\boldsymbol{x}) = \sum_{i \in B_2} \max_j \left\{ \sigma\left(\boldsymbol{w}^{(i)} \cdot \boldsymbol{x}_j^+\right) \right\}$$

$$N_{W_t}^{(2)}(\boldsymbol{x}) = -\sum_{i \in B_6} \max_j \left\{ \sigma\left(\boldsymbol{u}^{(i)} \cdot \boldsymbol{x}_j\right) \right\}$$

$$N_{W_t}^{(3)}(\boldsymbol{x}) = \sum_{i \in B_4} \max_j \left\{ \sigma\left(\boldsymbol{w}^{(i)} \cdot \boldsymbol{x}_j\right) \right\}$$
$$- \sum_{i \in B_8} \max_j \left\{ \sigma\left(\boldsymbol{u}^{(i)} \cdot \boldsymbol{x}_j\right) \right\}$$

and

$$N_{W_t}^{(4)}(\boldsymbol{x}) = N_{W_t}(\boldsymbol{x}) - N_{W_t}^{(1)}(\boldsymbol{x}) - N_{W_t}^{(2)}(\boldsymbol{x}) - N_{W_t}^{(3)}(\boldsymbol{x})$$
$$= \sum_{i \in B_1 \cup B_3} \max_j \left\{ \sigma\left(\boldsymbol{w}^{(i)} \cdot \boldsymbol{x}_j\right) \right\}$$
$$- \sum_{i \in B_5 \cup B_7} \max_j \left\{ \sigma\left(\boldsymbol{u}^{(i)} \cdot \boldsymbol{x}_j\right) \right\}$$

We would like to analyze the dynamics of $N_{W_t}(\boldsymbol{x}^+)$. To do so, we will address each $N_{W_t}^{(i)}(\boldsymbol{x}^+)$ in turn for $1 \le t \le t^*$.

**Bounding $N_{W_t}^{(1)}(\boldsymbol{x}^+)$:** By Lemma F.5 part 1, it follows that $N_{W_t}^{(1)}(\boldsymbol{x}^+) = \sum_{j \in B_2} \boldsymbol{w}_0^{(j)} + \frac{c_\eta t |B_2|}{2k}$ for all $1 \le t \le t^*$. Recall the definition of $\gamma$ in Eq. 14. Then, by Lemma F.2, for sufficiently small $\epsilon$ and $k > poly(\log d, \frac{1}{\epsilon})$ it holds that:

$$\left| \frac{|B_2|}{k} - \gamma \right| \le \epsilon$$

By Lemma F.4, we have:

$$\left| \sum_{j \in B_2} \boldsymbol{w}_0^{(j)} \right| \le |B_2| M \le (\gamma k + \epsilon k) M$$

Therefore, after $1 \le t \le t^*$ iterations, we have:

$$\left| N_{W_t}^{(1)}(\boldsymbol{x}^+) - \frac{c_\eta t \gamma}{2} \right| \le (\gamma k + \epsilon k) M + \frac{c_\eta t \epsilon}{2}$$

By choosing $M$ and $\epsilon$ to be sufficiently small (given an upper bound on $t$ that does not depend on $M$ and $\epsilon$, which we show later), $\left| N_{W_t}^{(1)}(\boldsymbol{x}^+) - \frac{c_\eta t \gamma}{2} \right|$ is sufficiently small.

**Calculating $N_{W_t}^{(2)}(\boldsymbol{x}^+)$:** Notice that after $n - 1$ iterations, we have by Lemma F.5 part 6 that $N_{W_t}^{(2)}(\boldsymbol{x}^+) = 0$. By taking $c_\eta$ to be sufficiently small, we can ensure that $n - 1 < t^*$.

**Bounding $N_{W_t}^{(3)}(\boldsymbol{x}^+)$:** By Lemma F.5 parts 4 and 8 and given that $M << c_\eta$ we have for $1 \le t \le t^*$:

$$N_{W_t}^{(3)}(\boldsymbol{x}^+) =$$

$$\frac{c_\eta}{2k} \left( \sum_{i \in \mathcal{P}_T \cap \mathcal{P}_s} |\mathcal{W}_0^+(i) \cap \mathcal{W}_0^-(2)| - \sum_{i \in \mathcal{P}_T \cap \mathcal{P}_s} |\mathcal{U}_0^-(i) \cap \mathcal{U}_0^+(1)| \right)$$

By Lemma F.3, for sufficiently small $\epsilon$, for any $i \in \mathcal{P}_T \cap \mathcal{P}_s$, the difference

$$\frac{1}{k} \left( |\mathcal{W}_0^+(i) \cap \mathcal{W}_0^-(2)| - |\mathcal{U}_0^-(i) \cap \mathcal{U}_0^+(1)| \right)$$

is sufficiently small. We conclude that $N_{W_t}^{(3)}(\boldsymbol{x}^+)$ is sufficiently small for small $\epsilon$.

**Bounding $N_{W_t}^{(4)}(\boldsymbol{x}^+)$:** By Lemma F.5, parts 1,3,5,7 it follows that

$$N_{W_t}^{(4)}(\boldsymbol{x}^+) \le kM$$

and thus can be made sufficiently small for small $M$.

**Finishing the proof:** By combining the previous arguments we have

$$N_{W_t}(\boldsymbol{x}^+) = \frac{c_\eta t \gamma}{2} + \beta^+$$

for $1 \le t \le t^*$, where $\beta^+$ is sufficiently small.

By symmetry, we have:

$$-N_{W_t}(\boldsymbol{x}^-) = \frac{c_\eta t \gamma}{2} + \beta^-$$

for $1 \le t \le t^*$, where $\beta^-$ is sufficiently small.

Our goal is to show that gradient converges to a global minimum at $T = t^*$. Let

$$t^+ = \underset{t \le t^*}{\arg\min}\left\{\frac{c_\eta t\gamma}{2} + \beta^+ > 1\right\}$$

and

$$t^- = \underset{t \le t^*}{\arg\min}\left\{\frac{c_\eta t\gamma}{2} + \beta^- > 1\right\}$$

where the minimum is over integral times $t$. Notice that $t^+! = t^-$ can only occur when there exists an integer $r$ such that

$$\left|1 - \frac{c_\eta r\gamma}{2}\right| \le 2\max\{\beta^+, \beta^-\} \tag{15}$$

Choose $c_\eta$ to be a small number which is not an integral multiple of $\frac{2}{\gamma}$ (e.g., choose irrational $c_\eta$). Then, $\max\{\beta^+, \beta^-\}$ can be made sufficiently small such that Eq. 15 does not hold. [12] In this case, after $\frac{1}{\gamma c_\eta} \le t^+ = t^- = t^* \le \frac{3}{\gamma c_\eta}$ iterations, gradient descent converges to a global minimum. $\qquad\square$

### F.3 FINISHING THE PROOF OF THEOREM 7.1

We are now ready to prove the theorem. Gradient descent converges to a global minimum after $\frac{1}{\alpha c_\eta} \le T \le \frac{3}{\alpha c_\eta}$ iterations by Lemma F.6. Furthermore, by the proof of Lemma F.6, $\mathcal{W}_T^+(1) = \mathcal{W}_0^+(1)$. For each $j \in \mathcal{W}_0^+(1)$, the norm of $\boldsymbol{w}_T^{(j)}$ is at least $\frac{\eta}{2}T \ge \frac{1}{2\gamma k}$. Therefore, for a sufficiently small $\epsilon$ by Lemma F.2:

$$M_{\boldsymbol{w}}^+(1) \ge \frac{|\mathcal{W}_0^+(1)|}{2\gamma k} \ge \frac{1}{3}$$

Now, by Lemma F.6, for all $j \notin \mathcal{W}_0^+(1)$, it holds that $\left|\boldsymbol{w}_T^{(j)}\right| \le \left|\boldsymbol{w}_0^{(j)}\right| + \frac{\eta}{2} \le \eta = \frac{c_\eta}{k}$. Therefore, for all $i \in \mathcal{P} \smallsetminus \{2\}$, $M_{\boldsymbol{w}}^-(i) \le c_\eta$. By symmetry, it follows that the PSI property holds with $b = 1$ and $c = \sqrt{18}c_\eta$.

## G EXPERIMENTAL DETAILS OF SECTION 8

Here we provide details of the experiments performed in Section 8. All experiments were run on NVidia Titan Xp GPUs with 12GB of memory. Training algorithms were implemented in PyTorch. All of the empirical results can be replicated in approximately one hour on a single Nvidia Titan Xp GPU.

We now describe how we created train and test sets for our setting. For train we sampled digits from the original MNIST training set and for test we sampled digits from the original MNIST test set. To sample a data point, we randomly sampled a label $y \in \{\pm 1\}$. Then, if $y = 1$ we randomly sampled 9 MNIST digits (either from the MNIST train or test set). Then randomly chose 8 of them to be the color green and one of them to be the color blue. If $y = -1$, we do the same procedure with blue replaced by red.

For training we implemented the setting in Section 3. Specifically, here we have $n = 9$ and $d = 28 * 28 * 3 = 2352$. We trained the network in Eq. 1 with $k = 20$ for training set sizes $m = 6$, $m = 20$ and $m = 1000$. For each training set size we performed 10 different experiments with different sampled training set and initialization. We ran SGD with batch size $\min\{10, m\}$, learning rate 0.0001 and for 200 epochs. We report the test accuracy and train accuracy in the final epoch (200). In all runs SGD gets 100% train accuracy. For $m = 6$ the mean test accuracy is 88.09% with standard deviation 12.7, for $m = 20$ the mean test accuracy is 99.84% with standard deviation 0.35 and for $m = 1000$ the mean test accuracy is 100% with standard deviation 0.

Figure 5, Figure 6 and Figure 7 show the set of all filters in the experiments reported in Figure 2c, for $m = 6$, $m = 20$ and $m = 1000$, respectively. To plot the figures, for each entry of the filter $x$

---

[12]Note that $\max\{\beta^+, \beta^-\}$ does not depend on $c_\eta$.

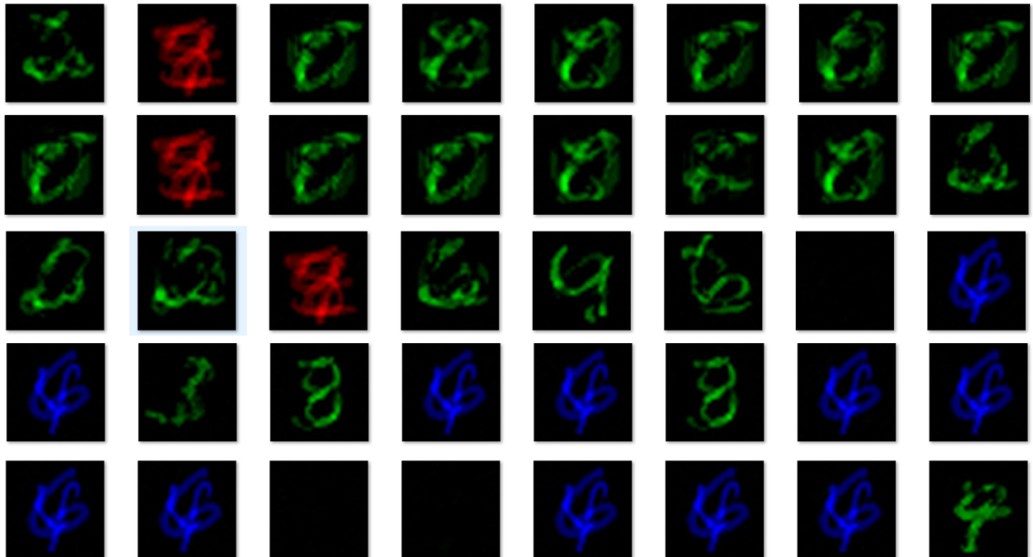

Figure 5: Learned filters for experiment with $m = 6$.

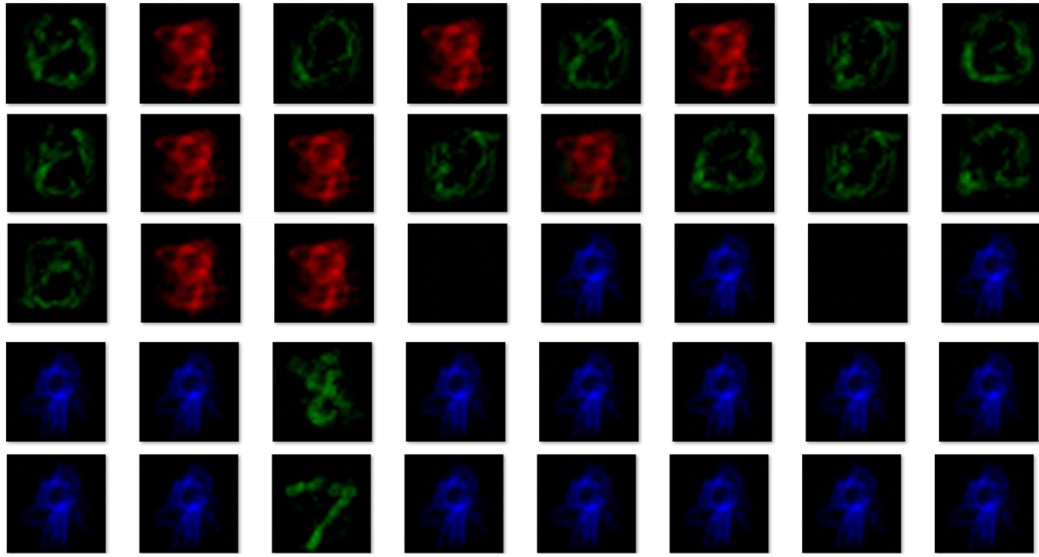

Figure 6: Learned filters for experiment with $m = 20$.

we calculated $\max\{0, x\}$. We scaled the weights of the network to be with values between 0 to 255 by dividing all entries by the maximum entry across all parameters of the network (after performing $\max\{0, x\}$) and multiplying by 255.

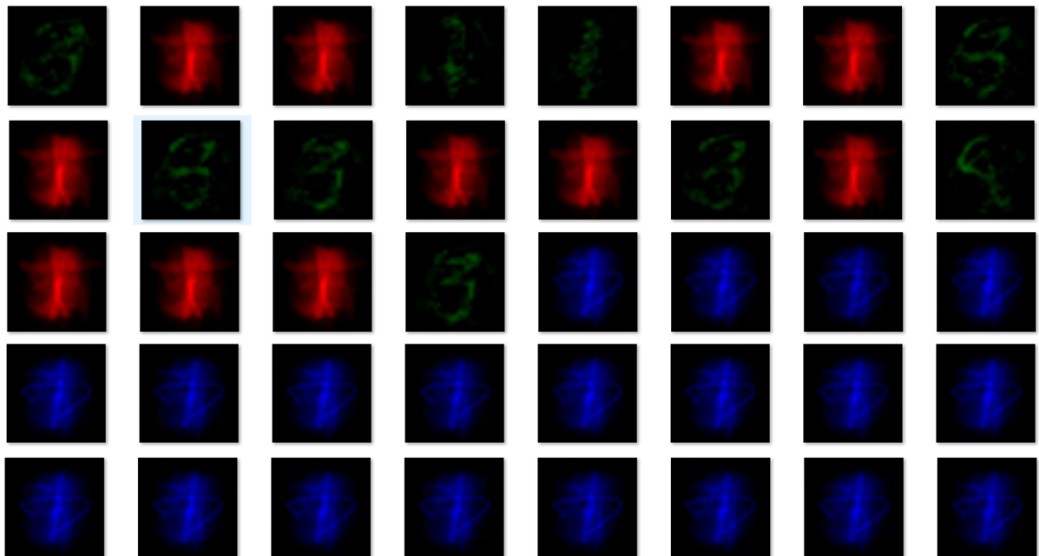

Figure 7: Learned filters for experiment with $m = 1000$.

