# OpenReview forum: "On the Inductive Bias of a CNN for Distributions with Orthogonal Patterns"
_ICLR.cc/2021/Conference — Reject_

### Official Review · AnonReviewer3 · 2020-10-17
**Interesting ideas on theory for CNN's**

**Rating:** 6
**Confidence:** 3

**Review:**

This paper is concerned with the question of generalization of convolutional neural networks. For that, the authors study a simple toy model, where each data point consists of several patterns. All patterns are assumed to be orthogonal to each other. Those images should be learned with a 3-layer neural network. The contributions of this paper are as follows:

1. The authors show that the networks, which are analyzed in this paper, have large VC dimension. In particular, this implies that generalization cannot be explained via generic VC theory.
2. As a way out, the authors introduce the concept of Patterns Statistics Inductive Bias (PSI). This relates the weight to the patterns in the dataset. In particular, the authors prove that PSI implies good generalization.
3. The authors study the conjecture, that training a neural network via gradient descent (with hinge loss) leads to a network, which satisfies the (PSI) condition.  They provide empirical as well as some theoretical evidence.

I think this is an interesting paper, which develops a new point of view and contains interesting ideas. While, the paper does not provide full evidence that the (PSI) condition is the right way to look at the problem, it may open up a new research direction.

On a personal level, I think point 1. and 2. are argued convincingly in the paper, but 3. could be strengthened with more evidence. For example, Theorem 7.1 allows for only two data points. It would be interesting to see a more general version of this theorem.

Remarks:
-I was wondering about Figure 1c). What happens, if you increase $\max (0, \frac{s_i}{s_1}) $ above $1$?
-Why do you choose $b=2$ for the experiments in Section 7?


Some typos:
-p. 6 Figure 1c): It should be $\max (0, \frac{s_i}{s_1}) $.

--------------------------------
I want to thank the authors for their rebuttal. I will stick to my decision and not change the score.

---

> ### Author Response · Authors · 2020-11-16
> **Proving a general theorem is a major challenge and we corroborate our theory with a large number of experiments.**
>
> Thank you for the positive feedback. We are happy that you find the paper interesting.
>
> ###
> “On a personal level, I think point 1. and 2. are argued convincingly in the paper, but 3. could be strengthened with more evidence. For example, Theorem 7.1 allows for only two data points. It would be interesting to see a more general version of this theorem.”
> ######
>
> We agree that a more general theorem would strengthen the results. However, the two-point analysis is already involved and it is a major challenge to analyze more general setups due to non-convexity. We note that our large number of experiments provide strong evidence that PSI holds in practice. Furthermore, using our analysis we get a very precise understanding of the inductive bias of GD on a variant of MNIST and why it has very low sample complexity. This is highly non-trivial and we believe this can set the ground to understand how overparameterized CNNs work on other vision tasks.
>
>
>
> ###
> Remarks: -I was wondering about Figure 1c). What happens, if you increase $\max(0,\frac{si}{s1})$ above
> 1? Why do you choose b=2  for the experiments in Section 7?
> Some typos: -p. 6 Figure 1c): It should be
> $\max(0,\frac{si}{s1})$.
> #####
>
> - In this case there are more negative points with pattern i than positive points (which all have pattern 1). It should be $\max(0,-\frac{si}{s1})$ and not $\max(0,\frac{si}{s1})$.
> - We chose $b=2$ to show that PSI holds with small constants.

---

### Official Review · AnonReviewer1 · 2020-10-19
**pattern statics inductive bias is an interesting idea but the theory is limited and requires unrealistic assumptions**

**Rating:** 5
**Confidence:** 5

**Review:**

This paper studied a simplified image classification task with orthogonal non-overlapping patches and is learned by a 3-layer CNN. The authors observed pattern statics inductive bias (PSI) in experiments. They proved that if a learning algorithm satisfies PSI, the sample complexity is nearly quadratic in the filter dimension; while the VC dimension of the network is at least exponential in the filter dimension. The authors also verified PSI in some task based on MNIST that has non-orthogonal patches.

My major concern about this paper is that the theory seems to only work under orthogonal patterns and non-overlapping filters, unfortunately, neither of which is true in practice. So it’s unclear if this theory can explain the success of CNN in practice. I also have other concerns:
1. This is mainly a theory paper, but the theoretical contribution is pretty limited. The theory only considers orthogonal patterns and non-overlapping filters. Theorem 6.1 needs to assume that PSI holds and spurious patterns are unbiased; under these assumptions, the result is pretty straightforward. This paper didn’t prove PSI holds for SGD except in a toy example (two training points).
2. The lower bound on the VC dimension is only exponential in the filter size. This lower bound is pretty weak since in practice the filter size is usually small (e.g. 3x3).
3. The experiment is run on an artificial task based on MNIST images. It would be much more convincing if the experiment is on standard tasks (like 10-class classification in MNIST). Since the theoretical setting is limited, it needs experiments to show how the theory applies to practical tasks.

---

> ### Author Response · Authors · 2020-11-16
> **The orthogonal distribution is useful for understanding generalization of CNNs and nonoverlap networks are studied and used in practice.**
>
> Thank you for your thoughtful review. We address your concerns below:
>
> ###
> “My major concern about this paper is that the theory seems to only work under orthogonal patterns and non-overlapping filters, unfortunately, neither of which is true in practice. So it’s unclear if this theory can explain the success of CNN in practice.”
> #####
>
> With regards to the orthogonality assumption, we believe it is a good starting point for analyzing overparameterized CNNs. The distribution captures the idea of inputs with specific patterns and that the pixels are correlated. This is in contrast to standard IID assumptions on coordinates such as isotropic Gaussian distributions which are assumed in many works.
>
>
> Regarding the nonoverlapping assumption, we note that non-overlapping filters are used in practice. Furthermore, many theoretical works have analyzed CNNs with non-overlapping filters due to their tractability and difficulty of analyzing overlapping filters. See "``On the Expressive Power of Overlapping Architectures of Deep Learning" of Sharir et al. for a survey. Thus, we believe it is instructive to make this assumption for analyzing CNNs.
>
>
> In Section 8, we use our analysis to understand overparameterized CNNs on a variant of MNIST. We get a good understanding of the inductive bias of GD in this task and why it has very low sample complexity. This is highly non-trivial and we believe this can set the ground to understand how overparamaterized CNNs work in other vision tasks.
>
> We note that our setting is significantly closer to practice than other published works which consider networks with *one* channel (e.g., Du et al., 2018) or *linear* CNNs (Gunasekar et al., 2018).
>
>
>
>
> ###
> “This is mainly a theory paper, but the theoretical contribution is pretty limited. The theory only considers orthogonal patterns and non-overlapping filters. Theorem 6.1 needs to assume that PSI holds and spurious patterns are unbiased; under these assumptions, the result is pretty straightforward. This paper didn’t prove PSI holds for SGD except in a toy example (two training points).”
> #####
>
>
> Analyzing the optimization and generalization of gradient methods on overparameterized CNNs is extremely challenging. Even for “simple”' settings (such as ours), the optimization is non-convex and providing a complete analysis of optimization and generalization is not tractable using current mathematical methods. Therefore, expectations regarding what can be achieved with theoretical analysis should be set accordingly.
>
> To make progress on this challenging task, we provided a proof in a simple setting and corroborated our findings in a large number of experiments. We also note that our VC dimension lower bound is a novel and non-trivial theoretical result.
>
>
> ###
> The lower bound on the VC dimension is only exponential in the filter size. This lower bound is pretty weak since in practice the filter size is usually small (e.g. 3x3).
> #####
>
> We note that there are cases where filter sizes are larger, e.g., in AlexNet the first layer has convolutions of size 11x11. More importantly, the VC result emphasizes the difference between solutions that gradient descent converges to and overfitting solutions that it in principle *can* converge to. The result shows that a network can have zero training error while only detecting the spurious patterns. This network is vastly different from the network that gradient descent converges to, which mainly detects the discriminative patterns.
>
>
> ###
> The experiment is run on an artificial task based on MNIST images. It would be much more convincing if the experiment is on standard tasks (like 10-class classification in MNIST). Since the theoretical setting is limited, it needs experiments to show how the theory applies to practical tasks.
> #####
>
> Although the variant of MNIST we consider is artificial, it does capture real-life data properties, such as different written digit patterns. Understanding what overparameterized CNNs learn in this setting and why they have low sample complexity is highly non-trivial. Using our analysis we provide novel insights that accurately predict the behavior of gradient descent and that better explain why it has low sample complexity. Therefore, we believe that this result is significant. Understanding why overparameterized CNNs generalize on MNIST is a major challenge  and we believe that our work sets the ground for tackling it.
>
> We hope that we have clarified the novelty of our results and our contribution to the understanding of overparameterized CNNs. We kindly ask you to reevaluate your score based on our comments.

---

### Official Review · AnonReviewer4 · 2020-10-28
**I vote for accept.**

**Rating:** 6
**Confidence:** 5

**Review:**

In this manuscript the authors derive theoretical analysis for the generalization guarantees of a naïve CNN (3-layers) where the task is a simplified binary classification task, under the assumption that the images contain orthogonal patches (a naïve assumption). They define a statistical phenomenon that holds in SGD in the proposed setting and call it Pattern Statistics Inductive Bias (PSI). Informally, this means that the magnitude of the dot-product between the learned pattern detectors and their detected patterns is correlated with the distribution of the patterns in the data.   They prove that if a learning algorithm  satisfies PSI then its sample complexity is O(d^2 log (d)), where d is the dimension of the filter. According to their empirical derivation SGD satisfies this property. In contrast there exist learning algorithms that have exponential sample complexity.

Pros.
1.	Addressing the problem of generalization guarantees is important and interesting.
2.	Novelty. In some sense, this works shows theoretical results which are less restrictive than Yu at el, circumventing the dependence of the sample complexity on the network size.
3.	Although the setting presented in this manuscript is quite limited, empirical observations on MNIST are in line with the analysis.

Cons.
1.	It is not easy / straightforward to follow the manuscript.
2.	Specifically, the intuition, given in Sec. 4, is not clear.
3.	The proposed setting, which assumes orthogonal patches in images, is very limited and not natural.

---

> ### Author Response · Authors · 2020-11-16
> **The orthogonal pattern assumption shares features with real images and our analysis is applicable in a non-orthogonal setting.**
>
> Thank you for the positive feedback.
>
> With regards to the orthogonality assumption,we believe it is a good starting point for analyzing overparameterized CNNs. First, the distribution captures the idea of inputs with specific patterns and that the pixels are correlated as in real image data. This is in contrast to standard IID assumptions on coordinates such as isotropic Gaussian distributions which are assumed in many works. Second, the analysis on orthogonal patterns distribution accurately predicts the behavior of gradient descent on a variant of MNIST which has non-orthogonal patterns. Therefore, the analysis is useful beyond the orthogonality assumption
>
> We clarified the intuition in section 4.

---

### Official Review · AnonReviewer2 · 2020-11-02
**Restricted model and complex measures make this work less attractive**

**Rating:** 5
**Confidence:** 3

**Review:**

This paper studies a new theoretical framework to understand the ability of
ConvNets to deal with pattern recognition tasks. The authors suggest a new property
of convents where filters have large dot products with patterns occurring in images classified as 1 and small dot products with patterns appearing with images classified as 0.
It is assumed that there are two unique patterns whose occurrence in an image determines if the image is classified by 1 or 0.

I like the classification problem studied in the paper and the attempt to understand how ConvNets work from first principles.

My issue with this paper is that it uses rather cumbersome measures (PSI and detection ratios). The setting is very specific (one type of architecture, realizability assumption, a single positive pattern and a single negative one).
The combination of these factors casts doubts on whether these measures and analysis will be of any use elsewhere.  The authors claim:

"Empirically, we identify a novel property of the solutions found by SGD. We observe that the statistics of patterns in the training data govern
the magnitude of the dot-product between learned pattern detectors and their detected patterns. Specifically, patterns that appear almost exclusively in one of the classes will have a large dot-product with the channels that detect them. On the other hand, patterns that appear roughly equally in both
classes, will have a low dot-product with their detecting channels. We formally define this as the “Pattern Statistics Inductive Bias” condition (PSI) and
provide empirical evidence that PSI holds across a large number of instances. We also prove that SGD indeed satisfies PSI in a simple setup of two points in the training set"

I find this somewhat unsatisfying. The novel property mentioned by the authors is one of the first that comes to mind when thinking about the success of ConvNets. It seems very
unlikely that this has not been observed before (at least empirically). While obtaining rigorous proofs of properties of NN is hard, one would expect that for the simple setting studied
by the authors there would be a simpler explanation for the so-called PSI.

This paper makes numerous restrictions and assumptions.
Some examples:

"a natural model in this context is a 3-layer network with a convolutional layer, followed by ReLU, max pooling and a fully-connected layer."
Why is this a natural architecture? Has it been studied before or used before?
Also in the classification task why is it assumed that n<d?

There are many more such examples.

The proof of Theorem 5.1 uses the Sherman-Morrison formula without giving the formula and showing how it is used.

The survey of related work is very short, not mentioning many relevant works looking into similar pattern recognition problems.

Some examples:
Learnable and Nonlearnable Visual Concepts (Shvayster, 1990)On learning visual concepts and DNF formulae (Kushilevitz, Roth, 1996)

---

> ### Author Response · Authors · 2020-11-16
> **We first need to study restricted models to make headway in understanding generalization of overparameterized CNNs**
>
> Thank you for your feedback. We address your comments below.
>
> ###
> “The novel property mentioned by the authors is one of the first that comes to mind when thinking about the success of ConvNets. It seems very unlikely that this has not been observed before (at least empirically). While obtaining rigorous proofs of properties of NN is hard, one would expect that for the simple setting studied by the authors there would be a simpler explanation for the so-called PSI.”
> #####
>
> We are not aware of any work which shows the PSI condition and that it implies good sample complexity. We think that it is unfair to say that there should be simple results in our setting without providing any references. As we discuss below, even though our setting may appear “simple”, it is highly non-trivial to analyze.
>
> ###
> “The setting is very specific (one type of architecture, realizability assumption, a single positive pattern and a single negative one).“
> #####
>
> We agree that our setting is specific, but the difficulty of the problem should be considered and expectations on which settings can be analyzed should be set accordingly.
>
> Studying the generalization of SGD on overparameterized CNNs is extremely challenging. To the best of our knowledge, there is no *single* setting where we have a good understanding of generalization in this case. Indeed, there are major difficulties in this research direction. First, even for “simple”' settings (such as ours), the optimization is non-convex and providing a complete analysis of optimization and generalization is not tractable using current mathematical methods. Second, defining mathematical properties of real-world data (such as images) is very challenging.
>
> To tackle this problem, we considered a novel setup which both shares features with practical architectures and real-world classification problems and can be analyzed using theoretical and empirical methods. We provide a detailed analysis of overparameterized CNNs in our setting and gain very good understanding of their generalization properties. We further show that we gain very precise insights into the inductive bias of SGD on a variant of MNIST (that contains non-orthogonal patches), which is highly non-trivial and novel.
>
>
> The distribution we assume captures the idea of inputs with specific patterns and that the pixels are correlated as in real image data. This is in contrast to IID assumptions assumed in many works (such as isotropic Gaussian distributions). Moreover, our architecture assumption is significantly closer to practice than other works which consider networks with *one* channel (e.g., Du et al., 2018) or *linear* CNNs (Gunasekar et al., 2018).
> Therefore, we believe our work is significant and is a valuable contribution to the understanding of learning overparameterized CNNs.
>
> ###
> “My issue with this paper is that it uses rather cumbersome measures (PSI and detection ratios).”
> #####
>
> Our main goal is to understand generalization. Of course, it is better if the measures used to understand generalization are very simple, but we believe that understanding generalization with any reasonable measure is significant. PSI involves algebraic notation of the network but conceptually it is not complicated. It bounds the ratio between the response to spurious pattern detectors and the response of discriminative pattern detectors. When this ratio is small, the network generalizes well (because it has low response to spurious patterns). The bound depends on the statistics of the patterns in the training set (e.g., if a spurious pattern appears roughly equally in both classes, the upper bound is small). Furthermore, in the MNIST experiments we provide an informal and simple definition of PSI and show that it empirically holds.
>
> ###
> “The combination of these factors casts doubts on whether these measures and analysis will be of any use elsewhere.“
> #####
>
> We disagree. In Section 8, we have shown that our analysis accurately predicts the inductive bias of SGD in a variant of MNIST with non-orthogonal patches. Therefore, we see that the analysis is applicable in other settings. We believe that the ideas in this work can be used to understand even more complex settings.
>
> Regarding other remarks:
>
> We clarified the proofs and cited the papers you mentioned. Both papers deal with learnability questions of visual patterns distributions. Of course, our distribution is learnable (e.g., by SVM). However, our focus is on learnability using a specific algorithm and architecture: SGD trained on overparameterized CNNs. This is similar to works that study networks in learnable settings (e.g., linear networks on linearly separable data in Gunasekar et al., 2018).
> We require n<d because we sample n patterns without replacement out of d patterns.
>
> We hope that we have clarified the novelty of our results and our contribution to the understanding of overparameterized CNNs. We kindly ask you to reevaluate your score based on our comments.

---

### Decision · Program_Chairs · 2021-01-07
**Final Decision**

**Decision:**

Reject

**Comment:**

This paper considers a new model of input data specific for image classification problems. In particular, the high level idea is that each image contains certain patterns, and which patterns it contain decides its label. In this framework, under some stronger assumptions (e.g., patterns are orthogonal, one positive pattern and one negative pattern, PSI assumption, etc.) the authors showed that SGD on a 3-layer overparametrized convolutional network will be able to have a small sample complexity, while the VC dimension would be at least exponential in d. The paper also provided some empirical evidence on a modified MNIST dataset. While the idea seems to be an interesting first step, the reviewers find that the current version of theory still relies on fairly strong assumptions.